# The peptidase DA1 cleaves and destabilizes WUSCHEL to control shoot apical meristem size

Guicai Cui[1,2,5], Yu Li[1,5], Leiying Zheng [2,3] ✉, Caroline Smith[4], Michael W. Bevan [4] & Yunhai Li [1,2] ✉

Stem cells in plants and animals are the source of new tissues and organs. In plants, stem cells are maintained in the central zone (CZ) of multicellular meristems, and large shoot meristems with an increased stem cell population hold promise for enhancing yield. The mobile homeodomain transcription factor WUSCHEL (WUS) is a central regulator of stem cell function in plant shoot meristems. Despite its central importance, the factors that directly modulate WUS protein stability have been a long-standing question. Here, we show that the peptidase DA1 physically interacts with and cleaves the WUS protein, leading to its destabilization. Furthermore, our results reveal that cytokinin signaling represses the level of DA1 protein in the shoot apical meristem, thereby increasing the accumulation of WUS protein. Consistent with these observations, loss of *DA1* function results in larger shoot apical meristems with an increased stem cell population and also influences cytokinin-induced enlargement of shoot apical meristem. Collectively, our findings uncover a previously unrecognized mechanism by which the repression of DA1 by cytokinin signaling stabilizes WUS, resulting in the enlarged shoot apical meristems with the increased stem cell number during plant growth and development.

Plants and animals have a similar stem cell microenvironment, although they have evolved independently of each other[1,2]. Unlike most animals, plants can generate new shoot and root organs throughout their lifespan by continuous production of pluripotent stem cells in specific structures or tissues called meristems[1,3]. The Shoot Apical Meristem (SAM) is the source of all post-embryonic cells in aerial parts of a plant[4]. The Arabidopsis SAM contains three distinct layers of cells (L1-L3), which differentiate into epidermis, ground tissues, and vascular tissues of stem, leaf, and flower, respectively[5]. The SAM can be divided into different functional domains according to its functions and cytological features. The region containing pluripotent stem cells is called the Central Zone (CZ), in which cells divide slowly and form Peripheral Zones (PZ). The Organizing Center (OC) is a group of cells beneath the CZ region, which specifically initiates and maintains the activity of stem cells[6–8]. Meristem size is determined by the number of stem cells, and these influence the size of organs derived from the meristem[9–11]. The homeodomain transcription factor WUSCHEL (WUS) has a central function in maintaining meristem stem cells[7]. *WUS* is specifically expressed in the OC, and the null mutations in *WUS* result in loss of stem cell properties and premature termination of the shoot apical meristem[12,13]. Conversely, induction of WUS in the central zone leads to increased apical meristem size[14]. *WUS* expression is inhibited by a signaling cascade involving the diffusible peptide CLAVATA3 (CLV3) that binds to the transmembrane receptor kinase

[1]Key Laboratory of Seed Innovation, Institute of Genetics and Developmental Biology, Chinese Academy of Sciences, Beijing, China. [2]College of Advanced Agriculture, University of Chinese Academy of Sciences, Beijing, China. [3]Key Laboratory of Plant Molecular Physiology, Institute of Botany, Chinese Academy of Sciences, Beijing, China. [4]John Innes Centre, Norwich Research Park, Norwich, UK. [5]These authors contributed equally: Guicai Cui, Yu Li. ✉e-mail: lyzheng@ibcas.ac.cn; yhli@genetics.ac.cn

CLV1 and receptor-like protein CLV2[15–18]. Loss of function of this cascade leads to larger meristems due to *WUS* over-expression[19–21]. WUS protein also moves from the OC to the CZ to activate the expression of *CLV3*[22,23]. Thus, they form a feedback loop that balances stem cell maintenance and cell differentiation in the shoot apical meristem[24]. WUS functions as a heterodimer with HAIRY MERISTEM (HAM) transcription factors to regulate the stem cell microenvironment by regulating common downstream genes[25]. WUS-HAM heterodimers confine *CLV3* expression to the outer apical meristem layer[26]. WUS also interacts with SHOOT MERISTEMLESS (STM) to regulate *CLV3* expression and control stem cell number[27]. In addition, several transcriptional regulators have been described to regulate shoot stem cell function by influencing *CLV-WUS* signaling in Arabidopsis[28–32]. Despite the central role of WUS in the initiation and maintenance of stem cell fate, modulation of WUS protein levels remains poorly understood.

The phytohormones cytokinin and auxin play important roles in the regulation of shoot meristems. Cytokinin signaling promotes the enlargement of the SAM and increases shoot stem cell number by activating the expression of the *WUS* gene[33]. In contrast, WUS can repress the transcription of several two-component *ARABIDOPSIS RESPONSE REGULATORS* genes (*ARRs*)[34]. Recently, cytokinin signaling has also been reported to stabilize WUS protein, resulting in the accumulation of WUS protein[29]. Therefore, cytokinin signaling promotes the accumulation of WUS protein through both transcriptional and post-translational regulation of WUS. However, how cytokinin signaling promotes WUS protein accumulation remains unclear due to the lack of an established mechanism for WUS degradation.

The cleavage of proteins is an important way to determine the functions of targeted proteins in plants and animals. We have previously shown that the peptidase DA1 is a central regulator of seed and organ size[35,36]. Here we show that the peptidase DA1 directly cleaves and destabilizes WUS protein in the SAM. Cytokinin signaling influences this process by repressing DA1 protein levels. Consequently, the *da1-1* mutant produces larger shoot apical meristems with an increased stem cell population, and DA1 activity is required for cytokinin-induced enlargement of shoot apical meristem. Our findings reveal an important genetic and molecular mechanism linking cleavage of WUS by DA1 to cytokinin signaling and apical meristem size control.

## Results

### DA1 regulates shoot stem cell function and SAM size

The peptidase DA1 limits seed and organ growth in Arabidopsis[35,36]. The *da1-1* and *da1-1^{Ler}* mutants formed larger seeds, leaves, inflorescences and flowers than their parental lines Col-0 and Ler, respectively (Supplementary Fig. 1)[35,36]. We also observed that sepal, petal, and stamen numbers and carpel sizes in *da1-1* and *da1-1^{Ler}* were increased compared with those in their parental lines (Supplementary Fig. 1). Enlarged inflorescences and an increased number of floral organs are associated with larger shoot meristems[37]. We therefore examined the shoot apical meristem (SAM) size of *da1-1^{Ler}* and *da1-1*. As shown in Supplementary Fig. 2, *da1-1^{Ler}* has a larger SAM and inflorescence meristem (IM) compared with Ler. Similarly, the SAM in *da1-1* was obviously larger than that in Col-0 (Fig. 1a, b). The quantification data also showed that the average area of the *da1-1* SAM was significantly increased compared with that of Col-0 (Fig. 1b and Supplementary Fig. 3). We previously demonstrated that the genomic fragment containing the *DA1* gene (*DA1COM*) complemented the large seed and organ phenotypes of *da1-1*[36]. The genomic fragment (*DA1COM*) also complemented the enlarged SAM phenotype of *da1-1* (Fig. 1a, b). A previous study showed that over-expression of *DA1* in *35 S:GFP-DA1* plants led to smaller organs[38]. Similarly, *35 S:GFP-DA1* seedlings also had smaller SAM than the wild type (Fig. 1a, b). Therefore, DA1 negatively regulates shoot meristem size. Expression of *DA1* in the SAM was assessed in a *pDA1:GUS* transgenic line[36]. As expected, *DA1* is strongly expressed in the SAM region (Fig. 1c), consistent with its role in defining SAM size. We also observed that *DA1* is strongly expressed in younger leaves (Fig. 1c and Supplementary Fig. 4), consistent with a previous report[36]. An increased number of cells expressing the stem cell marker gene *CLV3* were seen in the *da1-1* SAM compared to the wild type SAM (Fig. 1d, e), indicating that *da1-1* increases stem cell production in SAM. Thus, these results demonstrated that DA1 is involved in the regulation of shoot stem cell function and meristem size.

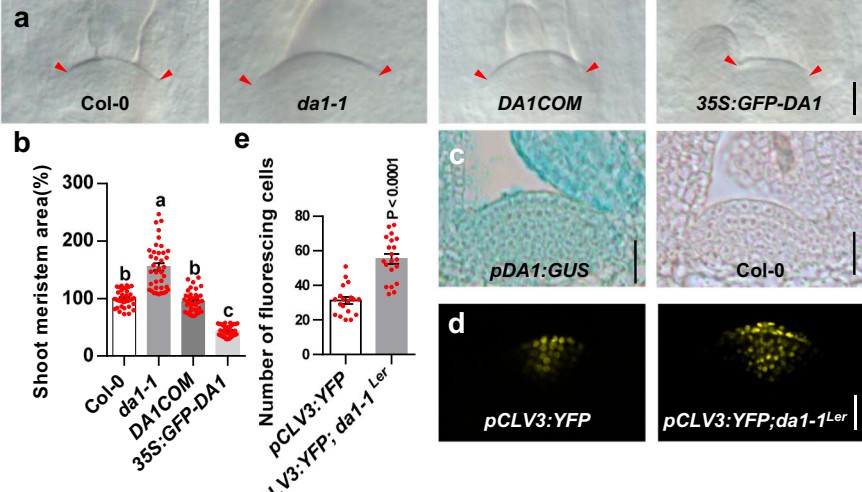

**Fig. 1 | The *da1-1* forms large shoot apical meristem with increased stem cell population. a** SAMs of Col-0, *da1-1*, *DA1COM* and *35S:GFP-DA1* (*n* = 38). DA1COM represents that the *da1-1* mutant was transformed with the DA1 genomic DNA. **b** The average SAM area of Col-0, *da1-1*, *DA1COM* and *35 S:GFP-DA1* (*n* = 38). The measured region of SAM was shown in Supplementary Fig. 3. Data are mean ± s.e.m. relative to the wild-type value (100%). One-way ANOVA with Tukey's multiple comparison test was used for statistical analyses(P < 0.05). **c** Images of SAMs from *pDA1:GUS* plants(*n* = 15). **d** Images of SAMs from *pCLV3:YFP* (*n* = 19)(left) and *pCLV3:YFP; da1-1^{Ler}* (*n* = 20)(right) plants. YFP (yellow) fluorescence was shown. **e** The number of fluorescing cells in SAMs of the *pCLV3:YFP* (*n* = 19) and *pCLV3:YFP; da1-1^{Ler}* (*n* = 20) plants. Data are mean ± s.e.m. relative to the *pCLV3:YFP* value. *P* values are from two-sided Student's *t* tests. Scale bars, 20 μm (**a**, **c**). 10 μm (**d**). All the plants were grown for 6 days in long-day conditions. The experiments were done with similar results in at least two independent replicates. Source data are provided as a Source Data file.

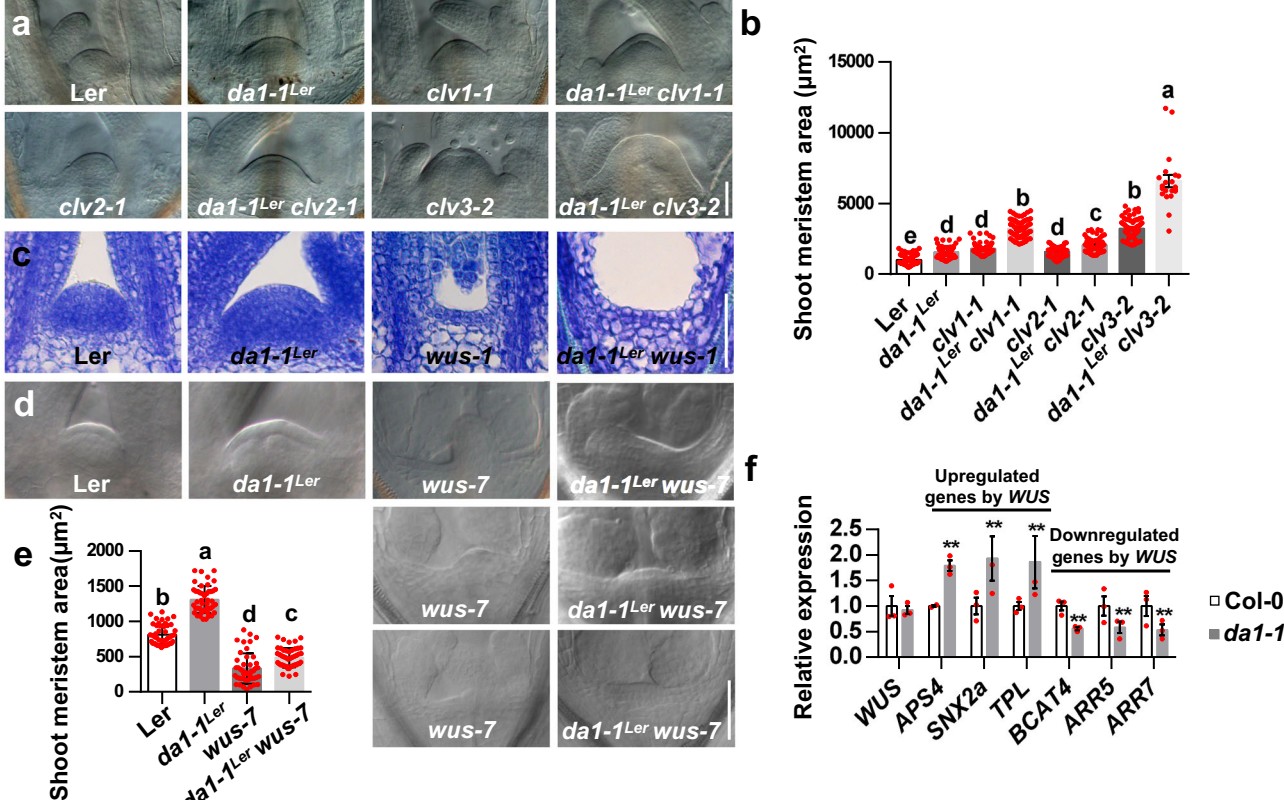

**Fig. 2 | DA1 acts genetically with WUS to control SAM size. a** SAMs of Ler (*n* = 55), *da1-1^Ler^*(*n* = 56), *clv1-1*(*n* = 63), *da1-1^Ler^ clv1-1*(*n* = 62), *clv2-1*(*n* = 47), *da1-1^Ler^ clv2-1*(*n* = 58), *clv3-2* (*n* = 63), and *da1-1^Ler^ clv3-2*(*n* = 21). **b** The average SAM area of Ler (*n* = 55), *da1-1^Ler^*(*n* = 56), *clv1-1*(*n* = 63), *da1-1^Ler^ clv1-1*(*n* = 62), *clv2-1*(*n* = 47), *da1-1^Ler^ clv2-1*(*n* = 58), *clv3-2* (*n* = 63), and *da1-1^Ler^ clv3-2*(*n* = 21). **c** SAMs of Ler, *da1-1^Ler^*, *wus-1* and *da1-1^Ler^ wus-1*(*n* = 20). **d** SAM of Ler, *da1-1^Ler^*, *wus-7* and *da1-1^Ler^ wus-7*(*n* = 46). **e** The average area of Ler, *da1-1^Ler^*, *wus-7*, and *da1-1^Ler^ wus-7*(*n* = 46) SAMs. **f** The relative expression levels of a set of genes, which have been reported to be upregulated or downregulated by the inducible activation of WUS. SAMs of 6-day-old Col-0 and *da1-1* were used to perform quantitative RT-PCR assay. Data was normalized with ACTIN2. Data are mean ± S.D (*n* = 3 for three biological repeats) relative to the wild-type value. P values are from two-sided Student's *t* tests. \*\**P* < 0.01 compared with the wild type (Col-0). Data in **b** and **e** are presented as mean values ± s.e.m. One-way ANOVA with Tukey's multiple comparison test was used for statistical analyses (*P* < 0.05). Scale bars, 50 μm (**a, c, d**). All the plants were grown for 6 days in long-day conditions. The experiments in **a, c, d** were done with similar results in at least two independent replicates. Source data are provided as a Source Data file.

## DA1 acts genetically with WUS to control shoot stem cell function and SAM size

The regulation of SAM size and stem cell population depends on a *WUS-CLV3* feedback loop[7,12,19–21]. SAMs of *clv* mutants are significantly enlarged, and the number of floral organs in *clv* mutants is increased, similar to SAM phenotypes observed in *da1-1* and *da1-1^Ler^* mutants. To explore genetic relationships between *DA1* and *CLV* genes in SAM size control, we generated *da1-1^Ler^ clv1-1*, *da1-1^Ler^ clv2-1* and *da1-1^Ler^ clv3-2* double mutants. As shown in Supplementary Fig. 5, *da1-1^Ler^* strongly enhanced the numbers of *clv1-1*, *clv2-1*, and *clv3-2* sepals, petals, stamens, and carpels, respectively, suggesting a synergistic genetic interaction between *DA1* and *CLV* genes in floral organ number control. SAM sizes of *clv1-1*, *clv2-1*, and *clv3-2* were dramatically enhanced by the *da1-1^Ler^* (Fig. 2a, b). These analyses indicated that the *da1-1^Ler^* mutation synergistically enhanced the large SAM phenotype of *clv1-1*, *clv2-1*, and *clv3-2* mutants, suggesting that *DA1* may act redundantly or in parallel with *CLV1*, *CLV2*, and *CLV3* to control SAM size and stem cell population[39].

These synergistic genetic interactions between *DA1* and the *CLV* pathway suggested that they may share a common downstream component[39]. The *CLV* pathway restricts expression of *WUS*, a central regulator of shoot stem cell proliferation[7,12,19–21]. To assess whether *WUS* mediates the effect of *DA1* on SAM size, we generated a *da1-1^Ler^ wus-1* double mutant. The *wus-1* mutant is not able to form a normal shoot apical meristem[12]. Similarly, the *da1-1^Ler^ wus-1* double mutant did not form a normal shoot apical meristem (Fig. 2c and Supplementary Fig. 6a). The *da1-1^Ler^ wus-1* double mutant exhibited similar plant growth phenotypes to *wus-1* (Fig. 2c and Supplementary Fig. 6a, b). We also observed that the SAM region of the *da1-1^Ler^ wus-1* double mutant was obviously wider than that of the *wus-1* single mutant (Fig. 2c), suggesting that DA1 and WUS may have partially overlapped function in SAM size control. As DA1 has several diverse substrates[35,40,41], it is plausible that some of its substrates might be involved in the regulation of SAM size. Considering that *wus-1* is a very strong allele and can not form SAM, we further used the weak *wus-7* allele to conduct additional genetic analyses. The *wus-7* seedlings formed small distorted shoot apical meristems compared to the wild type (Fig. 2d). The *da1-1^Ler^ wus-7* double mutant had an overall similar SAM shape to the *wus-7* single mutant (Fig. 2d), but the quantification of SAM size showed the *wus-7* mutation partially suppressed the large SAM phenotype of *da1-1^Ler^* (Fig. 2d, e). As *wus-7* is not a null allele, it is possible that any remaining activity of WUS in *wus-7* may still be influenced by *da1-1*. Thus, it seems reasonable that the *da1-1^Ler^ wus-7* double mutant had slightly larger SAM than *wus-7*. These genetic analyses indicated that the large SAM of *da1-1* partially depends on the functional *WUS*. Quantitative RT-PCR results revealed a common set of genes that are regulated by inducible activation of *WUS* and by *da1-1* (Fig. 2f)[42], while *WUS* expression was not affected by *da1-1* mutation (Fig. 2f). These results provided further evidence for possible common roles of *DA1* and *WUS* in regulating shoot stem cell function.

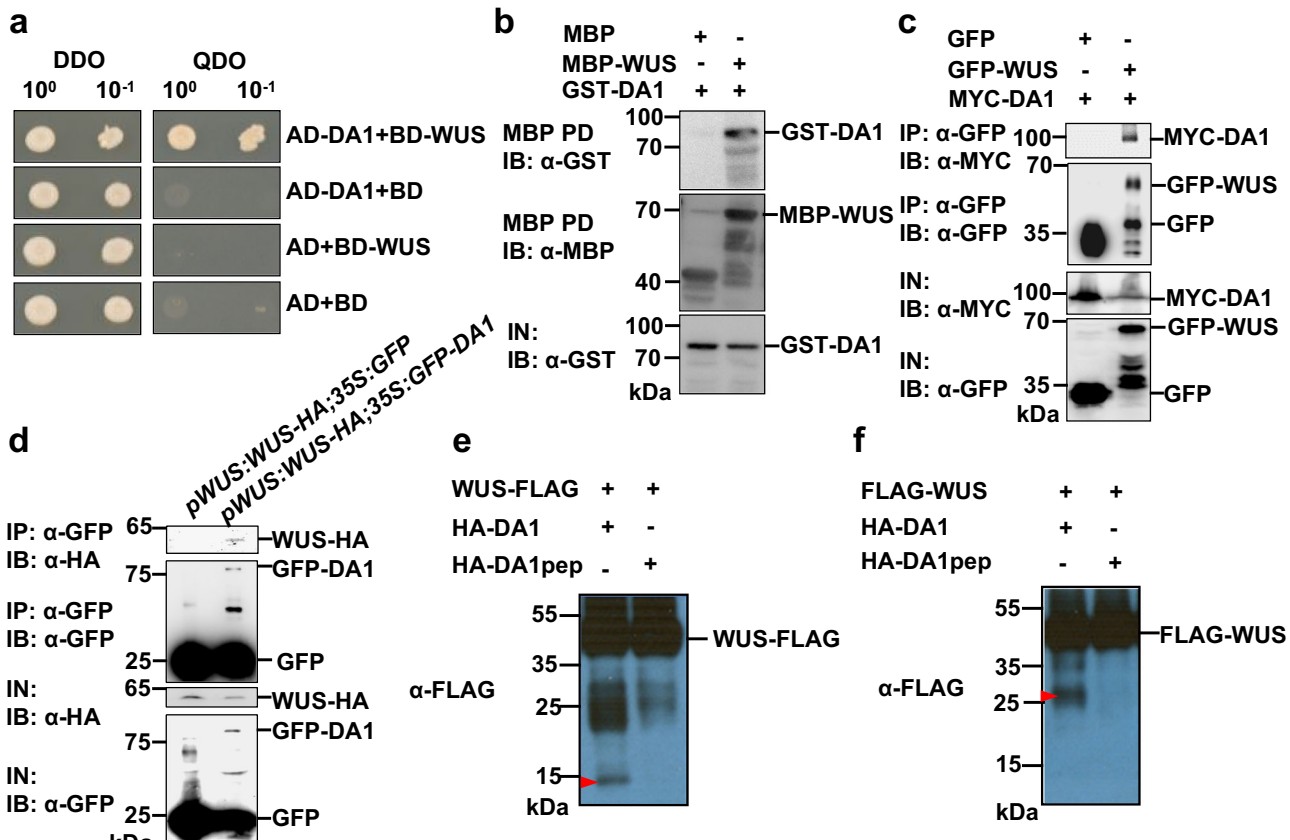

**Fig. 3 | DA1 physically interacts with and cleaves WUS. a** The indicated construct pairs were co-transformed into yeast strain Y2H Gold (Clontech). Interactions were examined on the control media DDO (SD/-Leu/-Trp) and selective media QDO (SD/-Ade/-His/-Leu/-Trp). **b** DA1 directly interacts with WUS in vitro. GST-DA1 was pulled down (PD) by MBP-WUS immobilized on amylose resin and analyzed by immunoblotting (IB) with an anti-GST antibody. **c** WUS interacts with DA1 in vivo. *N. benthamiana* leaves were transformed by injection of *Agrobacterium tumefaciens* GV3101 cells harboring *35S:GFP-WUS* and *35S:MYC-DA1* plasmids. MYC-DA1 was detected in the immunoprecipitated GFP-WUS complex, indicating that there is a physical association between DA1 and WUS in vivo. **d** Co-immunoprecipitation

analyses showing the interactions between WUS and DA1 in Arabidopsis. Total protein extracts of *pWUS:WUS-HA; 35S:GFP* and *pWUS:WUS-HA; 35S:GFP-DA1* plants were incubated with GFP-Trap agarose beads. Precipitates were detected by western blot with anti-GFP and anti-HA antibody. **e**, **f** Arabidopsis *da1-ko1 dar1-1* mesophyll protoplasts were cotransformed with plasmids expressing *WUS-FLAG* or *FLAG-WUS* with *HA-DA1* and *HA-DA1pep*, respectively. The specific cleavage products of WUS-FLAG and FLAG-WUS were indicated by red arrowheads. IP immunoprecipitation; IN input; IB immunoblot. Experiments in **b**–**f** were repeated independently at least twice with similar results. Source data are provided as a Source Data file.

## DA1 physically interacts with and cleaves WUS

As *DA1* acts genetically with *WUS*, and *WUS* expression was not changed in the *da1-1* mutant, we asked whether DA1 could directly interact with WUS. Yeast two hybrid assays showed that DA1 interacted with WUS in yeast cells (Fig. 3a). The WUS protein comprises a homeodomain, a dimerization domain, the WUS-box and the EARL motif, we then asked which domain of WUS could interact with DA1. Surprisingly, we found that most of deletion versions of WUS fused to BD autoactivated the reporter gene (Supplementary Fig. 7a). In addition, the full-length DA1 or the UIM domains fused to BD also autoactivated the reporter gene (Supplementary Fig. 7b), consistent with a previous report[41]. We therefore used DA1-LIM + C (LIM domains and the peptidase domain) fused to BD to test the interactions between DA1 and the domains of WUS, respectively. Interestingly, we found that the DA1-LIM + C interacted with the C terminal region of WUS (the dimerization domain, WUS-box and EARL motif), although the DA1-LIM + C did not interact with any single domain of WUS (Supplementary Fig. 7c), respectively. We further found that WUS did interact with AD-DA1-LIM in yeast cells (Supplementary Fig. 7d).

To further confirm the interaction between DA1 and WUS, we performed in vitro pull-down assays. GST-DA1 and MBP-WUS fusion proteins were expressed in *E.coli*, respectively. Pull-down assays showed that GST-DA1 can bind MBP-WUS, but not the negative control

MBP (Fig. 3b). The interaction between DA1 and WUS was then examined in planta by transient expression of *35S:GFP-WUS* together with *35S:MYC-DA1* in *N. benthamiana* leaves. The co-immunoprecipitation (Co-IP) analysis showed that MYC-DA1 was detected in the immunoprecipitation complex of GFP-WUS, but not in that of the negative control GFP (Fig. 3c), indicating that DA1 associates with WUS in vivo. We further investigated whether DA1 associated with WUS in Arabidopsis. We crossed *35S:GFP-DA1* transgenic line with *pWUS:WUS-HA* and generated *35S:GFP-DA1; pWUS:WUS-HA* lines. Co-immunoprecipitation (Co-IP) assay revealed that WUS-HA was immunoprecipted with GFP-DA1, but not with the GFP only (Fig. 3d), indicating that DA1 associates with WUS in Arabidopsis. The additional bands below the full-length GFP-WUS or GFP-DA1 in Co-IP assays might be the degraded products as observed in several previous studies (Fig. 3c, d)[35,40,43]. Finally, bimolecular fluorescence complementation (BiFC) assays in *N. benthamiana* leaves also showed that DA1 associates with WUS in nuclei (Supplementary Fig. 8). Taken together, these results revealed that DA1 interacts with WUS in vitro and in vivo.

DA1 is a metallopeptidase that cleaves diverse growth regulatory proteins[35]. As DA1 genetically and physically interacts with WUS, we tested whether DA1 could cleave WUS by transient co-expression in protoplasts[35]. When WUS-FLAG was transiently co-expressed with HA-

DA1, but not HA-DA1pep (no peptidase activity)[35], the specific cleavage product of WUS-FLAG was detected (Fig. 3e and Supplementary Fig. 9a). When we put the FLAG tag in the N terminus of WUS (FLAG-WUS), similar results were obtained (Fig. 3f and Supplementary Fig. 9b). Thus, these results indicated that the peptidase DA1 can cleave WUS.

## DA1 destabilizes WUS

DA1 peptidase cleaves its substrates and causes their destabilization[35]. Because our results showed that DA1 physically interacts with and cleaves WUS, this prompted us to investigate the stability of WUS protein in *da1-1* plants. In transgenic *pWUS:WUS-GFP* plants, GFP was observed in the OC of SAM and migrated into the CZ (Fig. 4a), consistent with previous reports[22,23]. In *pWUS:WUS-GFP; da1-1* plants, GFP florescence was observed in the OC and the CZ of the SAM (Fig. 4b). Cells with WUS-GFP were significantly increased in the SAM of *pWUS:WUS-GFP; da1-1* compared with those in the SAM of *pWUS:WUS-GFP* (Fig. 4a, b, e). Similar results were also observed in inflorescence meristems (IMs) of *pWUS:WUS-GFP; da1-1* (Fig. 4c, d, f). These results indicated that SAMs and IMs of *da1-1* plants contain more WUS proteins than those of the wild type.

The large SAM of *da1-1* may contain more cells with WUS expression. To distinguish between a general increase in WUS-expressing cells in the larger SAM of *da1-1* plants and the direct influence of DA1 activity on WUS protein levels within the SAM, we generated transgenic lines of the estradiol-inducible form of DA1 (*pER8:MYC-DA1*) and then crossed with *pWUS:WUS-GFP* to obtain *pWUS:WUS-GFP; pER8:MYC-DA1* plants. The *pWUS:WUS-GFP; pER8:MYC-DA1* seedlings treated with β-estradiol for 12 h led to the strong accumulation of MYC-DA1 protein (Fig. 4k). Conversely, cells with WUS-GFP in the SAMs of *pWUS:WUS-GFP; pER8:MYC-DA1* were dramatically decreased at 12 h after induction (Fig. 4g, h, l). By contrast, cells with WUS-GFP in the SAMs of *pWUS:WUS-GFP* (a negative control) was not changed at 12 h after β-estradiol treatment (Fig. 4i, j, m). The few cells with WUS-GFP in the SAMs of *pWUS:WUS-GFP; pER8:MYC-DA1* after β-estradiol treatment were not caused by the reduction of *WUS-GFP* gene expression, as the levels of *WUS* mRNA were similar in plants with or without β-estradiol induction (Fig. 4n, o). Considering that the β-estradiol treatment for 12 h did not obviously affect SAM size (Fig. 4g-j), the decreased cells with WUS-GFP in *pWUS:WUS-GFP; pER8:MYC-DA1* SAM after β-estradiol induction were not caused by SAM size. Thus, DA1 activity destabilizes WUS proteins in the Arabidopsis SAM.

## Specific expression of *DA1* driven by the *WUS* promoter represses the large SAM phenotype of *da1-1*

*WUS* is expressed in the OC, and WUS protein moves from the OC to neighboring stem cells. *DA1* is expressed in the entire shoot apical meristem. These observations suggested that DA1 may cleave and destabilize WUS in the OC and neighboring cells. Therefore, specific expression of *DA1* using the *WUS* promoter may, at least in part, rescue the large SAM phenotype of *da1-1*. As predicted, the shoot apical meristems of transgenic *da1-1* seedlings expressing *DA1* driven by the *WUS* promoter were smaller than those of *da1-1* (Fig. 5a-d, i). Surprisingly, we occasionally observed that transgenic plants exhibited deformed SAMs (Fig. 5e, f), like those observed in the *wus-7* mutant. It is possible that the cleavage of WUS by DA1 in the OC strongly reduces the stability of WUS in these transgenic plants, resulting in a *wus* mutant-like phenotype. Consistent with this, specific expression of *DA1pep* driven by the *WUS* promoter did not repress the large SAM phenotype of *da1-1* and also did not form distorted SAMs (Fig. 5g-i). These results further established a direct functional relationship between DA1 activity and WUS function.

## DA1 is involved in cytokinin-induced accumulation of WUS protein and enlargement of the SAM

Cytokinin signaling has been shown to activate the transcription of the *WUS* gene and increase the stability of WUS protein through post-translational regulation[29,33,34]. Nonetheless, it remains unclear how cytokinin stabilizes WUS protein due to the lack of an established mechanism for WUS degradation and cleavage. Given that DA1 can cleave and destabilize WUS, we tested whether DA1 is involved in the regulation of cytokinin-promoting stability of WUS protein. We first investigated the effect of cytokinin on DA1 protein level using *35S:GFP-DA1* transgenic plants. We treated *35S:GFP-DA1* plants with or without 6-benzylaminopurine (6-BA) for 8 h because treatment with 6-BA for 8 h did not obviously change the SAM size (Supplementary Fig. 10). Compared with mock-treatment, exogenous application of 6-BA significantly reduced the abundance of GFP-DA1 protein (Fig. 6a, b and Supplementary Fig. 11), while 6-BA did not affect mRNA levels of *GFP-DA1* in *35 S:GFP-DA1* (Fig. 6c), indicating that cytokinin signaling can repress DA1 protein levels. We also examined transcription of the *DA1* gene in Col-0 when treated with 6-BA for 8 h. Expression level of the *DA1* gene in 6-BA treated samples was similar to that in untreated samples (Fig. 6d). These results indicated that cytokinin signaling represses the accumulation of DA1 protein. It is possible that cytokinin represses the accumulation of DA1 protein in the SAM, in turn decreasing WUS cleavage and increasing its accumulation. To test this, we treated *pWUS:WUS-GFP* seedlings with 6-BA and observed cells with WUS-GFP in the SAMs. When treated with 6-BA for 8 h, cells with WUS-GFP in *pWUS:WUS-GFP* SAMs were strongly increased, whereas cells with WUS-GFP in the SAMs of *pWUS:WUS-GFP; da1-1* were only slightly increased (Fig. 6e-h, m). By contrast, the transcripts of *WUS-GFP* in *pWUS:WUS-GFP* and *pWUS:WUS-GFP; da1-1* plants were induced by 6-BA in a similar manner (Fig. 6n). Considering that 6-BA treatment for 8 h did not significantly influence the SAM size (Fig. 6e-h and Supplementary Fig. 10), cytokinin-induced accumulation of WUS protein involves DA1 activity.

Cytokinin induces the enlargement of the SAM[33,44,45]. To test whether DA1 is involved in this enlargement, we treated Col-0 and *da1-1* seedlings with 6-BA for three days and measured their SAM sizes. The 6-BA treatment increased the SAM size of Col-0 by 43.6%, while the SAM size was increased by 23.5% in *da1-1* (Fig. 6i-l, o). This indicated that the reduced DA1 function is partially required for cytokinin-promoting SAM enlargement. Consistent with our results, a previous study showed that cytokinin signaling may also promote the enlargement of SAM partially by influencing mRNA level of *WUS*[28]. Taken together, these results revealed that cytokinin promotes SAM enlargement by inducing the expression of the *WUS* gene and increasing the stability of the WUS protein partially by repressing DA1 accumulation.

## Discussion

Plant organs originate from the populations of stem cells in the meristems during post-embryonic development. WUS protein is a central regulator in the maintenance of stem cells in the shoot meristem. The regulation of protein stability is crucial for stem cell function in animals[46], but how WUS protein levels are modulated to control plant stem cell function remains unclear. We have previously shown that the peptidase DA1 limits seed and organ growth in Arabidopsis[36]. In this study, we demonstrated that DA1 physically interacts with and cleaves WUS (Fig. 3), causing its destabilization. Mutations in the peptidase domain of DA1 disrupted the cleavage to WUS, indicating that the peptidase activity of DA1 is required for WUS stability (Fig. 3e, f). The *da1-1* seedlings have large shoot apical meristems with an increased stem cell number (Fig. 1). Our biochemical, genetic and gene expression analyses support the conclusion that DA1 and WUS function, at least in part, in a common pathway to control the sizes of shoot apical meristem and stem cell

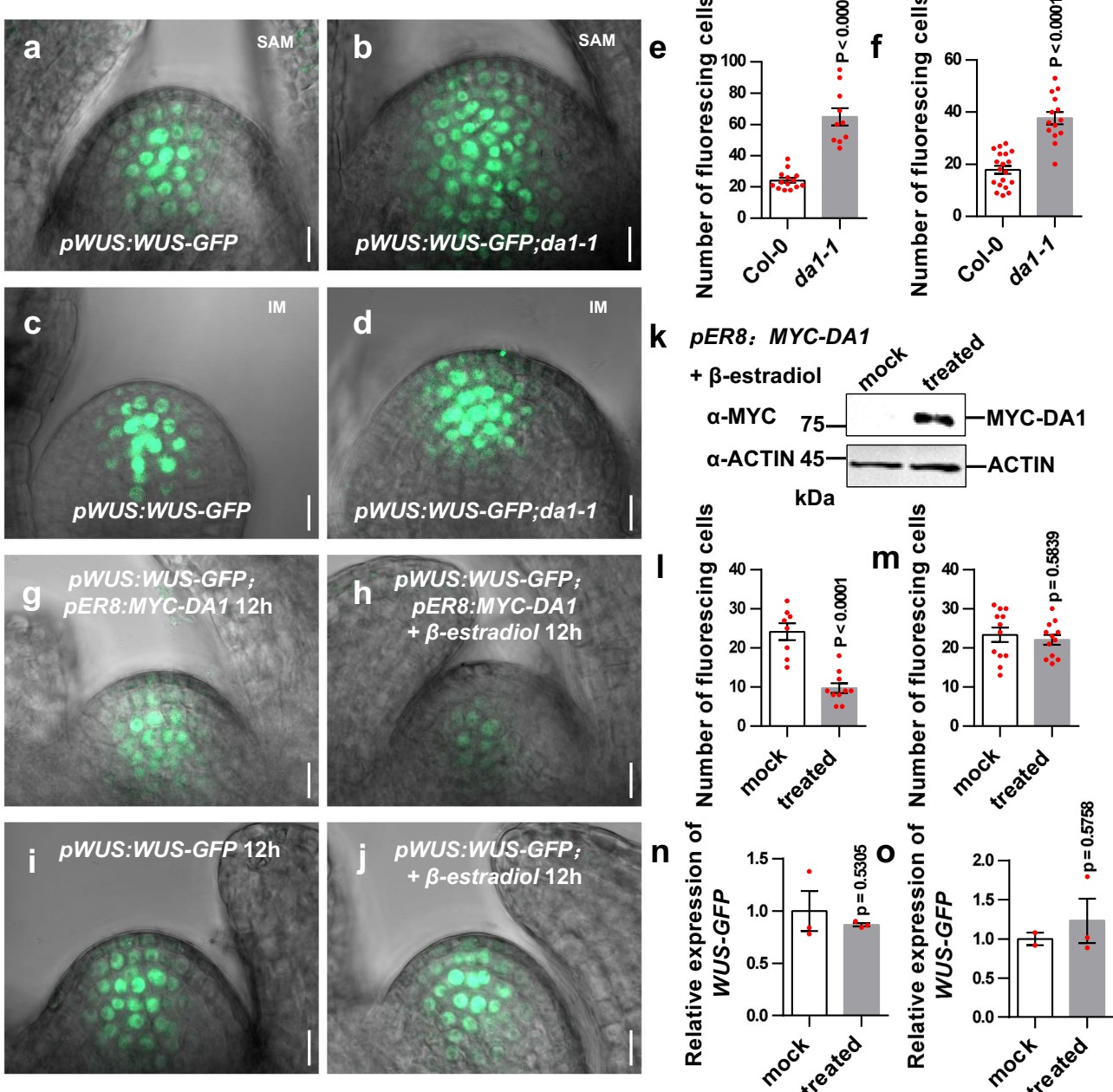

**Fig. 4 | DA1 destabilizes WUS. a**, **b** SAMs from *pWUS:WUS-GFP* (a)(*n* = 14) and *pWUS:WUS-GFP; da1-1* (b)(*n* = 10) transgenic plants. Plants were grown for 9 days in long-day conditions. **c**, **d** IMs from *pWUS:WUS-GFP* (**c**) (*n* = 19) and *pWUS:WUS-GFP; da1-1* (**d**) (*n* = 14) transgenic plants. Plants were grown for 35 days in long-day conditions. **e** The fluorescing cell number of SAMs from *pWUS:WUS-GFP* (*n* = 14) and *pWUS:WUS-GFP; da1-1* (*n* = 10) plants. **f** The fluorescing cell number of IMs in *pWUS:WUS-GFP* (*n* = 19) and *pWUS:WUS-GFP; da1-1* (*n* = 14) plants. **g**, **h** SAMs from *pWUS:WUS-GFP; pER8:MYC-DA1* plants treated without β -estradiol for 12 h (mock) (*n* = 8) and with β-estradiol for 12 h (treated) (*n* = 10). **i**, **j** SAMs from *pWUS:WUS-GFP* treated without β -estradiol for 12 h (mock) (*n* = 12) and with β -estradiol for 12 h (treated) (*n* = 12). **k** The protein levels of MYC-DA1 in *pWUS:WUS-GFP; pER8:MYC-DA1* transgenic plants treated with or without (mock) β -estradiol for 12 h. Total protein extracts were subjected to immunoblot assays using anti-MYC, and anti-ACTIN (as loading control) antibodies. **l** The fluorescing cell number of SAMs from *pWUS:WUS-GFP; pER8:MYC-DA1* plants treated without β-estradiol for 12 h (mock) (*n* = 8) and with β-estradiol for 12 h (*n* = 10). **m** The fluorescing cell number of SAMs from *pWUS:WUS-GFP* plants treated without β -estradiol for 12 h (mock) (*n* = 12) and with β -estradiol 12 h (treated) (*n* = 12). **n** Quantification of *WUS-GFP* mRNA levels in *pWUS:WUS-GFP; pER8:MYC-DA1* plants without β -estradiol for 12 h (mock) and with β -estradiol for 12 h (treated). Data are mean ± s.e.m with three biological replicates. **o** Quantification of *WUS-GFP* mRNA levels in pWUS:WUS-GFP plants treated without β -estradiol for 12 h (mock) and with β -estradiol for 12 h (treated). Expression level of *WUS-GFP* at mock was set at 1. Data are mean ± s.e.m with three biological replicates. Data in **e**, **f**, **l**, and **m** are presented as mean values ± s.e.m. *P* values are from two-sided Student's *t* tests. GFP (green) fluorescence were shown in **a–d**, **g–j**. Scale bars, 10 μm (**a–d**, **g–j**). Experiments in **a–d**, **g–k** were repeated independently at least twice with similar results. Source data are provided as a Source Data file.

populations (Fig. 2c–e). Although the *WUS* gene is expressed in the OC, WUS protein can move to neighboring cells to control stem cell function. Our results showed that *DA1* is expressed in the whole SAM (Fig. 1c), indicating that DA1 and WUS share overlapped expression domains. Similarly, several other factors that are not specifically expressed in stem cell regions play key roles in stem cell maintenance[47–49]. In addition, specific expression of *DA1* driven by the *WUS* promoter strongly represses the large SAM phenotype of

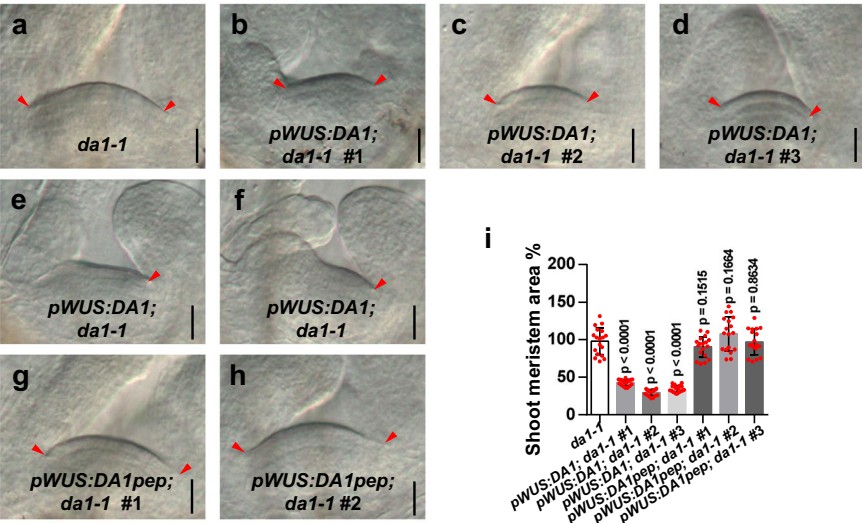

**Fig. 5 | Specific expression of DA1 driven by the WUS promoter represses the large SAM phenotype of *da1-1*. a** Morphologies of *da1-1* SAMs. **b**–**d** Morphologies of *pWUS:DA1; da1-1* SAMs. **e**, **f** Morphologies of abnormal SAMs in *pWUS:DA1; da1-1*. **g**, **h** Morphologies of *pWUS:DA1pep; da1-1* SAMs(*n* = 18). **i** The average SAM area of *da1-1, pWUS:DA1; da1-1* and *pWUS:DA1pep; da1-1* (*n* = 18). Data are mean ± s.e.m relative to the *da1-1* value (100%). One-way ANOVA with Dunnett's multiple comparison test was used for statistical analyses ($P < 0.05$). Scale bars, 20 μm (**a**–**h**). All plants were grown for 6 days in long-day conditions. The experiments in **a**–**h** were done with similar results in at least two independent replicates. Source data are provided as a Source Data file.

*da1-1* (Fig. 5), further supporting their close relationship. Thus, our findings reveal a previously unknown mechanism that the peptidase DA1-mediated cleavage and destabilization of WUS regulates shoot meristem size and shoot stem cell population. Considering that homologs of DA1 and WUS exist in different plant species, it is possible that utilization of DA1 and WUS homologs in different plant species could be a common theme in the regulation of shoot meristem size and plant growth and development. For example, repression of poplar vascular cambial cell division by PagDA1 involves promoting PagWOX4 degradation[50]. Our results show this likely involves PagWOX4 cleavage by DA1 and subsequent degradation.

Phytohormones are crucial for the maintenance of shoot stem cell homeostasis. Cytokinin signaling induces the transcription of the *WUS* gene and promotes the stability of WUS protein in the SAM[29,33,34]. Nonetheless, the underlying mechanism by which cytokinin stabilizes the WUS protein is still unknown due to the lack of an established mechanism for the cleavage and degradation of WUS protein. In this study, we found that cytokinin signaling represses the abundance of DA1 protein in the SAM (Fig. 6a, b and Supplementary Fig. 11). As DA1 can cleave and destabilize WUS, it is possible that cytokinin signaling promotes the accumulation of WUS protein partially by repressing DA1 protein. Supporting this notion, exogenous cytokinin treatment experiments showed that DA1 is involved in the regulation of cytokinin-induced accumulation of WUS protein and enlargement of the SAM (Fig. 6). Therefore, our findings define a regulatory mechanism in which the cleavage of WUS by DA1 connects cytokinin signaling and shoot stem cell number and meristem size regulation in Arabidopsis (Fig. 6p). However, we cannot rule out the possibility that CK could also affect WUS stability through another unknown mechanism as CK treatment can still induce the accumulation of the WUS protein and the enlargement of the SAM in *da1-1*, although the ratio of induction in *da1-1* was less than that in Col-0 (Fig. 6). In addition, it remains unclear how cytokinin signaling regulates DA1 protein levels. Considering that the phosphorylation of DA1 influences its stability[51], it will be a worthwhile challenge to investigate whether cytokinin receptors could interact with and phosphorylate DA1 in the future. A previous study showed that light and metabolic signals are able to activate the expression of the *WUS* gene, which also likely involves cytokinin signaling[52]. It will be interesting to examine whether the cytokinin-DA1-WUS module could mediate environmental cues (light and nutrients) to regulate stem cell function.

Larger meristems with an increased stem cell population hold promise for increasing yield in different plant species[9–11]. Our findings place DA1 at the crucial position of organ size, meristem size, and stem cell population regulation. Given that mutations in DA1 increase seed size in several crops[53–55], the regulation of fundamental stem cell proliferation by the DA1-WUS module has the potential to improve crop yield.

## Methods
### Plant materials and growth conditions
All the plant materials were used in Col-0 or Ler background as indicated. The *da1-1, da1-1^Ler, da1-ko1 dar1-1, DA1COM, 35S:GFP-DA1, wus-7* and *pWUS:WUS-GFP* lines were used in previous studies[35,36,41,49,56–58]. The seeds of *clv1-1* (NW45), *clv2-1* (NW46), *clv3-2* (N8066) and *wus-1* (N15), were obtained from the Nottingham Arabidopsis Stock Centre (NASC). The *da1-1^Ler clv1-1, da1-1^Ler clv2-1, da1-1^Ler clv3-2, da1-1^Ler wus-1* and *da1-1^Ler wus-7* double mutants were generated by crossing *da1-1^Ler* with *clv1-1, clv2-1, clv3-2, wus-1* and *wus-7*, respectively. The *pWUS:WUS-GFP; da1-1* was generated by crossing *da1-1* with *pWUS:WUS-GFP*. The *pWUS:WUS-GFP; MYC-DA1-PER8* was generated by crossing *pWUS:WUS-GFP* with *MYC-DA1-PER8*. All these mutants and double mutants were identified by allele-specific genotyping, transgenic antibiotic resistance screening or GFP signal detection. The primers used above were listed in Supplementary Data 1.

Seeds were sterilized in 100% isopropanol for 1 min and then 5%(v/v) sodium hypochlorite for 10 min. After sufficiently washed with sterile water, seeds were dispersed on 1/2 MS medium containing 1% (w/v) Suc with 0.9% agar and placed at 4°C for 3d. For long-day condition, plants were grown in controlled environmental chamber or in green house with 16 h light /8 h dark. For β-estradiol treatment, plants were grown for 7 days in 1/2MS solid medium and transferred to 1/2MS solid medium with 20 μM β-estradiol for 12 hours.

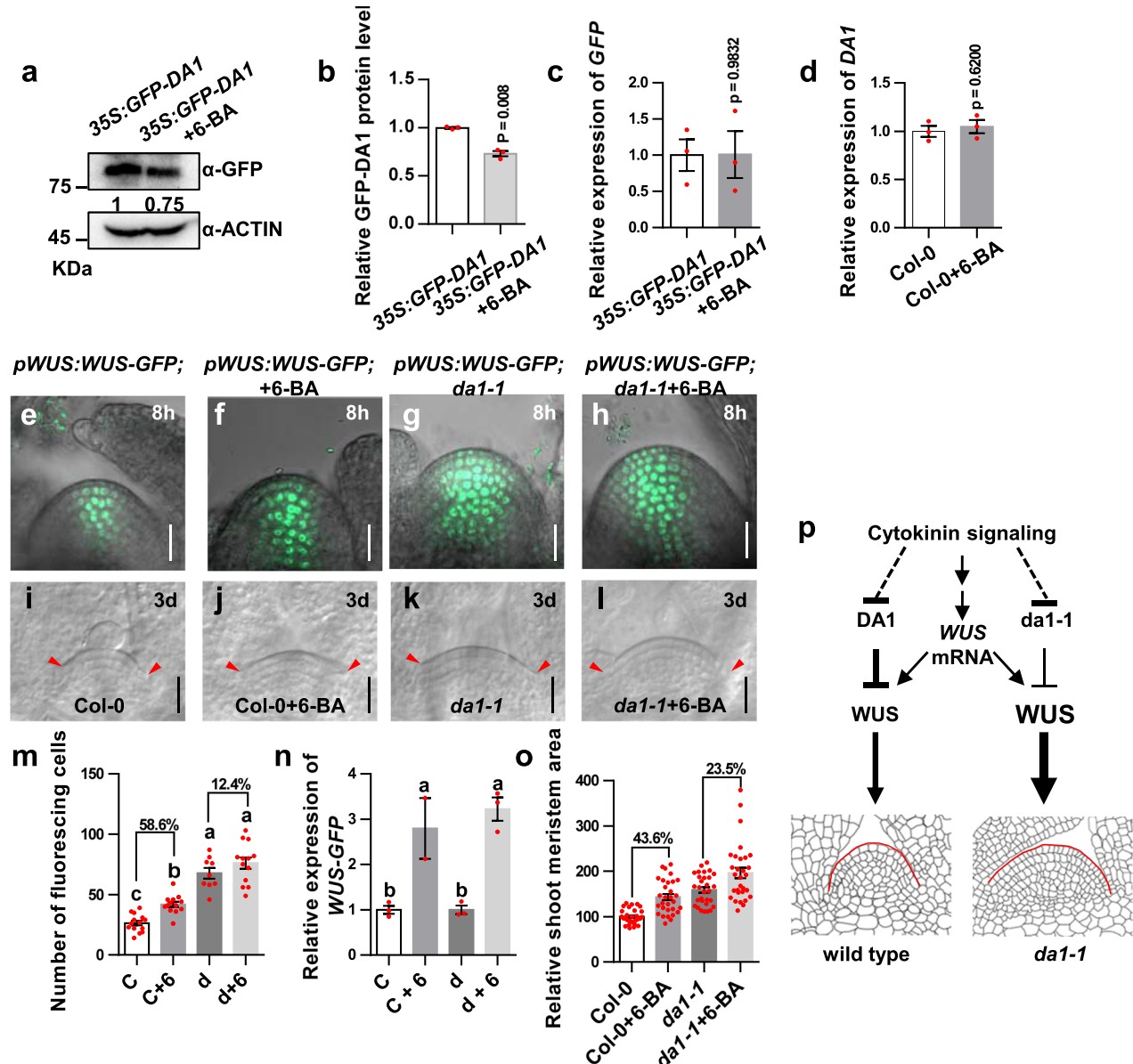

**Fig. 6 | DA1 is involved in cytokinin-induced accumulation of WUS protein and enlargement of the SAM. a**, **b** 6-BA treatment decreases the level of DA1 protein. **a** Total protein extracts were detected by anti-GFP and anti-ACTIN (loading control) antibodies **b** Quantification of GFP-DA1 protein levels in a. **c**, **d** Quantification of mRNA levels in *35 S:GFP-DA1* or Col-0 treated with or without 6-BA. **e** SAMs from *pWUS:WUS-GFP* transgenic plants ($n = 14$). **f** SAMs from *pWUS:WUS-GFP* transgenic plants treated with 6-BA ($n = 13$). **g** SAMs from *pWUS:WUS-GFP; da1-1* transgenic plants($n = 9$). **h** SAMs from *pWUS:WUS-GFP; da1-1* transgenic plants treated with 6-BA ($n = 13$). **i** SAMs of 6-day-old Col-0 plants. **j** SAMs of 6-day-old Col-0 plants treated with 6-BA. **k** SAMs of 6-day-old *da1-1* plants. **l** SAMs of 6-day-old *da1-1* plants treated with 6-BA ($n = 30$). **m** The fluorescing cell number of SAMs in *pWUS:WUS-GFP* ($n = 14$), *pWUS:WUS-GFP + 6-BA* ($n = 13$), *pWUS:WUS-GFP; da1-1* ($n = 9$), and *pWUS:WUS-GFP; da1-1+6-BA* ($n = 13$) plants. **n** Quantification of *WUS-GFP* mRNA levels. *pWUS:WUS-GFP* (C), *pWUS:WUS-GFP+6-BA* (C+6), *pWUS:WUS-GFP; da1-1* (d) and *pWUS:WUS-GFP; da1-1 + 6-BA* (d+6). **o** The average area of SAMs

of Col-0, Col-0 + 6-BA, *da1-1* and *da1-1* + 6-BA SAMs ($n = 30$). **p** A model for cytokinin-DA1-WUS module in controlling the shoot stem cell function and SAM size. DA1 represses WUS to regulate shoot stem cell function and SAM size, while Cytokinin represses DA1 level in the SAM. Cytokinin promotes the WUS accumulation partially through the repression of DA1. The *da1-1* mutation caused the WUS accumulation, resulting in the large SAMs with increased stem cell population. Cytokinin also promotes *WUS* expression. Data in **b**, **c**, **d** and **n** are mean ± s.e.m with three biological repeats. *P* values are from two-sided Student's *t* tests(**b**–**d**) and one-way ANOVA with Tukey's multiple comparison test (**n**). Data in **m** and **o** are presented as mean values ± s.e.m relative to the mock value (100%). *P* values are from one-way ANOVA with Tukey's multiple comparison test ($P < 0.05$). GFP (green) fluorescence were shown. Scale bars, 20 μm (**e**–**l**). The experiments in **e**–**m** and **o** were repeated independently at least twice with similar results. Source data are provided as a Source Data file.

## Confocal microscope observation

For SAM fluorescence signals detection, the shoot apices were detached and prepared from *pWUS:WUS-GFP*, *pWUS:WUS-GFP; da1-1*, and *pWUS:WUS-GFP; MYC-DA1-PER8* grown under long day condition (16 h light/ 8 h dark) according to a previous study[7]. Samples were observed using a Zeiss LSM 980 confocal microscopy, and the images used for fluorescencing cells counting were taken by meristem

confocal sections at suitable location. All the cell numbers with fluorescence signaling were then quantified by Image J.

## Differential interference contrast microscope

For shoot meristem size observation, 6-d-old and 9-d-old plants were harvested and cleared in clearing solution (Chloral hydrate 80 g, water 30 ml, Glycerol 10 ml) for 12 hrs. Samples were observed with DIC

microscope (Leica DM2500, Germany) and photographed using cooled CCD digital imaging system (Olympus BH2, Japan).

## SAM size measurement

For SAM size measurement, the measurement range takes L1 layer as the boundary and the junction of SAM and primordia on both sides as the starting points. The details can be found in Supplementary Fig. 3. SAMs were measured using Image J.

## Scanning electron microscopy

Samples were fixed in FAA solution with vacuum treatment for 30 min. Fixed samples were dehydrated with a gradual ethanol as 70%, 85%, 95%, and 100% (v/ v), then dried by critical-point drying (Hitachi HCP-2), and subsequently coated with a gold layer. The sputter-coated samples were ready for observation and imaging using scanning electron microscope (Hitachi S-3000N).

## Quantitative real-time PCR analysis

6-d-old aboveground part of Col-0 and *da1-1* were ground in liquid nitrogen, and the RNAprep pure kit was used to extract total RNA following the manufacturer's instructions (TIANGEN, Cat. No: DP439-H). cDNAs were synthesized using the FastQuant RT Super Mix kit (TIANGEN, Cat. No: KR108). qPCR reactions were assayed on a Lightcycler 480 machine (Roche Applied Science, USA) using 2×SYBR Green supermix kit (Vazyme, China). All individual reactions were done by three biological replicates, and the data were normalized to the *ACTIN2* gene. All primers were listed in Supplementary Data 1.

## Yeast two-hybrid assay

The Matchmaker Gold Yeast Two-Hybrid System was employed to perform the yeast two-hybrid assay in yeast strain AH109. *AD-DA1* and *DA1-LIM + C-BD* were used in previous studies[4]. The CDSs of *WUS* amplified using primers WUS-BD-F/R (Supplementary Data 1) was inserted into pGBKT7 vector (NcoI and PstI digestion) to get *BD-WUS* constructs using GBclonart Seamless Clone Kit (GB2001-48, Genebank Biosciences). The CDSs of WUS-HD, WUS-DD, WUS-C and WUS-C2 were inserted into pGADT7 vector (ECOR1 and SAC1 digestion) to get AD-WUS-HD, AD-WUS-DD, AD-WUS-C, and AD-WUS-C2 constructs. Primers are listed in the Supplementary Data 1. The Different combinations of plasmids as indicated were co-transformed in AH109 and grown on SD/-Trp/-Leu plates for 3 days. The protein interactions were then selected by spotting the yeast on SD medium with minus Trp/Leu/His/Ade and grown for 3 or more days.

## In vivo co-immunoprecipitation

The genomic sequence of *WUS* was inserted into *PMDC32* (in which the 35 S promotor was cleaved) to generate *pWUS:WUS-HA*. The construct primers are listed in the supplemental Data 1. The respective transgenic Arabidopsis plants of *35 S:GFP-DA1* and *pWUS:WUS-HA* were obtained by agrobacterium tumefaciens-mediated transformation. *pWUS:WUS-HA*; *35S:GFP-DA1*; and *pWUS:WUS-HA*; *35S:GFP* plants came from crossing *pWUS-WUS-HA* with *35S:GFP-DA1* and *35S:GFP*, respectively.

Aboveground part of *pWUS:WUS-HA;35 S:GFP-DA1*, and *pWUS:-WUS-HA;35S:GFP* 10-d-old plants were ground in liquid nitrogen. Total proteins were isolated with buffer (50 mM Tris-HCl, pH 7.5, 150 mM NaCl, 1% Triton X-100, 5% glycerol, 1 mM EDTA, 1× Roche protease inhibitor cocktail) and incubated with GFP-Trap®_A agarose beads (Chromotek, Cat. No: gta-20) for 1 h with agitation at 4 °C. Beads were washed four times with wash buffer (50 mM Tris [pH 7.5], 150 mM NaCl, 10% glycerol, 0.1% TritonX-100, 1 mM EDTA, and protease inhibitor cocktail). After adding 1×SDS loading buffer, the beads were heated at 98 °C for 5 min, and the corresponding proteins were subjected to sepatate in a 10% or 8% (w/ v) SDS-polyacrylamide gel, and detected with anti-GFP antibody (Abmart, Cat. No: M20004, dilution,

1:5000) and anti-HA antibody (Cwbio, Cat. No: 01239/34220, dilution, 1:5000).

## In vitro Pull-down assay

All the constructs were made by infusion cloning kit (GBclonart Seamless Clone Kit, GB2001-48, Genebank Biosciences). The CDS of *WUS* was infused into *pMALC2-MBP* digested with SalI and HindIII to generate the *MBP-WUS*. The CDSs of *DA1* was cloned into the *pGEX4T-1* vector digested with ECORI and SALI to generate the *GST-DA1*. Plasmids were transferred into *E. coli* BL21 (DE3) cells to expression the tag fused proteins. All proteins were induced with 0.4 mM Isopropyl β -D-1-thiogalactopyranoside (IPTG) at different conditions depending on proteins. The bacterial cells were then resuspended with TGH buffer (50 mM HEPES (pH 7.5), 1 mM EGTA, 150 mM NaCl, 1% (v/v) Triton X-100 and 10%(v/v) glycerol). The bacterial suspensions were sonicated on ice for 3 min at 20 amplitudes and centrifuged at 12,000×*g* for 10 min. The different combinations of MBP-WUS and GST-DA1 proteins with MBP agarose beads (NEB, Cat. No: E8037s) were incubated for 1 h at 4 °C with agitation, respectively. Beads were washed 5 times with washing buffer (50 mM HEPES (pH 7.5), 1 mM EGTA, 150 mM NaCl, 0.5% (v/v) TritonX-100, 1 mM PMSF and 10% (v/v) glycerol). After adding 1×SDS-loading buffer, the beads were heated 5 min at 98 °C and separated by SDS-PAGE gel. The immunoprecipitates were detected by GST antibody (Abmart, Cat. No: M20007M, 1:5000) and MBP antibody (NEB, Cat. No: #E8032, 1:10,000).

## Bimolecular fluorescence complementation assay

The CDS of *DA1* was amplified by specific primers YN-DA1-F/R, fused with the N-terminal fragment of YFP (nYFP), and subcloned into the linearized *pGWB414* vector (digested with XbaI and SalI) using in-fusion enzyme (Genebank Biosciences). The CDS of *WUS* was amplified by specific primers YC-WUS-F/R, fused with the C-terminal fragment of YFP (cYFP), and subcloned into the linearized *pGWB414* vector (digested with XbaI and SalI) using in-fusion enzyme (Genebank Biosciences). Primers used are listed in the supplemental information (Supplementary Data 1). Different combinations of Agrobacterium GV3101 containing the above plasmids were cotransformed into *N. benthamiana* leaves. After 48 h, YFP fluorescence was observed in leaves using an LSM710 confocal laser scanning microscope (Zeiss)

## Cleavage of WUS

The *35 S:HA-DA1* and *35 S:HA-DA1pep* constructs were used in previous studies[35,59]. The cDNA sequences of *WUS* was inserted into *pW1211* and *pW1266* to generate *35 S:WUS-FLAG* and *35 S:FLAG-WUS* by gateway cloning (Invitrogen), respectively. Primers are listed in the Supplementary Data 1.

The rosettes of *da1-kol dar1-1* were harvested before bolting to prepare protoplast. The ~0.5 mm fragments of leaves were incubated in enzyme solution (0.3% Macerozyme R-10, 10 mM $CaCl_2$, 0.4 M mannitol, 1.25% Cellulose RS, 20 mM MES at pH 5.7, 5 mM β-Mercatoethanol and 0.1% BSA) for 4 hrs in dark with gentle agitation. After digestion, the W5 solution (154 mM NaCl, 5 mM KCl, 2 mM MES pH 5.7 and 125 mM $CaCl_2$) was added, and followed 10–30 s with strongly shaking. By filtering through 40 µm nylon mesh with three to five washes using W5 solution, the protoplasts were collected by centrifugation at 100 × g for 5 min. The protoplasts were washed twice with W5 buffer and resuspended in MMG solution (0.4 M mannitol, 15 mM $MgCl_2$, and 4 mM MES pH 5.7) around 2× 10$^6$ cells mL$^{-1}$. The *35S:HA-DA1*, *35S:HA-DA1pep*, *35S:WUS-FLAG*, and *35S:FLAG-WUS* plasmids were extracted with Plasmid Maxprep Kit (Vigorous, Cat. No: N001). The combinations of constructs (10 µg) were added into 300 µL protoplasts with freshly prepared 330 µL polyethylene glycol (PEG) solution (0.1 M $CaCl_2$, 0.4 M mannitol, 40% w/v PEG4000), and incubated for 20 min on bench. The transfected protoplasts were then resuspended in W5 solution and incubated for 12-16 hrs at 28 °C in the

dark. Total proteins were isolated with extraction buffer (50 mM Tris-HCl pH 7.5, 150 mM NaCl, 0.1% Triton, 0.2% NP-40, 5 mM EDTA, and protease inhibitor cocktail). Proteins were detected by western blots with antibody against FLAG (Abmart, Cat. No: M20008, 1:5000) and HA (Cwbio, Cat. No: 01239/34220, dilution, 1:5000).

## The effect of 6-BA treatment on meristem size

For 6-benzylaminopurine (6-BA) treatment, 100 μM stock solutions of 6-BA (Sigma-Aldrich) in ddH$_2$O were dissolved in 1/2MS medium containing 1% (w/v) Suc with 0.9% agar prewarmed at 50 °C to a final concentration of 100 nM. For SAM size measurement, seedlings were grown on 1/2MS solid medium in long-day conditions (16 h light and 8 h dark at 22 °C) for 3 days, then transferred to 1/2MS solid medium with 100 nM 6-BA for 3 days or 8.67 days after seed stratification and then transferred to 1/2MS solid medium with 100 nM 6-BA for 8 h.

## Statistics analysis

All data are shown as the mean ± s.e.m. unless indicated otherwise. Statistical analysis was performed using GraphPad Prism 7 software (GraphPad Software, Inc. San Diego, CA, USA). All details on statistics have been indicated in figure legends. The exact P-value was included in the figures. No statistical method was used to predetermine sample size. Images were analyzed with ImageJ.

## Reporting summary

Further information on research design is available in the Nature Portfolio Reporting Summary linked to this article.

## Data availability

All materials in this study are available from the corresponding author upon request. The authors declare that all data supporting the findings of this study are available within the article and its Supplementary Information files. Arabidopsis reference genome (TAIR10) was used in this study. Source data are provided with this paper.

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

## Acknowledgements

We would like to thank Drs. Thomas Laux, Chuanyou Li, Yuxin Hu, Zhong Zhao, Zhaojun Ding, and Xigang Liu for mutant and transgenic seeds. We also thank NASC for *wus-1*, *clv1-1*, *clv2-1* and *clv3-2* seeds. We thank Dr. Yuling Jiao for helping WUS-GFP observation. This work was supported by the grants from the National Natural Science Foundation of China (32370357, 31872663 and 31961133001) and the strategic priority research program of the Chinese Academy of Sciences (XDB27010102).

## Author contributions

YH.Li conceived and designed this project. GC.Cui and YU.Li performed most experiments. C.S performed the cleavage of WUS. LY.Zheng, YU.Li, GC.Cui, M.B, and YH.Li analyzed data and wrote the manuscript.

## Competing interests

The authors declare no competing interests.
