## [Peer Review File · Nature Communications]

REVIEWER COMMENTS

Reviewer #1 (Remarks to the Author):

The manuscript from Cui and colleagues unravels a new mechanism of regulation of the key meristematic gene WUS which involves DA1, a peptidase previously reported as an organ size regulator. The work of the authors supports that 1. DA1 regulates stem cell activity in the SAM, 2. DA1 physically interacts with WUS and cleaves it, 3. DA1 is involved in post-transcriptional regulation of WUS by CK.

Overall, I find this study very interesting as it not only provides a new mechanism of regulation of WUS but also identifies a new player involved in the post-transcriptional regulation of WUS by CK. However, I wonder if the authors could not go even further to test the functional relevance of DA1-dependent regulation of WUS by CK proposed in their final model, as several articles have showed that WUS levels can be modulated in response to various stresses through CK. The article from Pfeiffer and colleagues (<https://elifesciences.org/articles/17023>) notably shows that light and metabolic signals can influence WUS levels through CK. Have the authors tried to look at the response of WUS to these signals in their lines where DA1 was misregulated? Moreover, the data from Fig.6 also supports that CK regulates DA1 in a post-transcriptional manner. Could the author discuss (or even better provide data), on the mechanisms through which DA1 could be regulated by CK?

Regarding the experiments presented in the paper, I have issues that would need to be addressed:

- I find most of the westerns blots of Fig. 3 not very clear. In the panel b and c, what are the additional bands we see, notably when WUS and DA1 are together? Are they product of degradation of WUS or non-specific binding? In panel 3d, why is there such a strong band corresponding to GFP alone in the right lane when GFP-DA1 is expressed? Is DA1-GFP cleaved? And what is the intermediate line between the GFP and the DA1-GFP? In panels e and f, we can hardly see the bands. Given the quality of the blots presented (especially Fig.3e and Fig.3f), I think that additional replicates are needed (notably independent experiments, see further comment on reproducibility). I also think that Nature Communications requires to show the raw blots, that I could not find.

- The article provides data supporting that the effect of DA1 on WUS is not transcriptional (especially Fig. 4c-i). However, altering WUS activity will affect SAM size, which should lead to changes in the number of WUS-expressing cells, which will ultimately influence WUS expression and protein levels (in a kind of positive feedback loop). I think that the authors should consider that point when interpreting the results of Fig. 2d (expression of WUS targets in DA1 mutants) and that they should provide data on the expression of WUS in the da1 mutant together with the data shown in Fig.4 (by qPCR or/and through the analysis of a transcriptional reporter for WUS).

- The data from the CK treatments in Fig.6 are quite important for the model presented in Fig.6h. If I understood correctly the material and methods, they were obtained by performing CK treatment for 7 days, which is a rather long time. I understand that few days are needed to see an effect of CK on SAM size (although 7 days is still a lot) but this is not the case for WUS as the data from Fig. 4 supports that DA1 can induce changes in WUS protein levels in less than 8h. Given my previous comment and the fact that long-term CK treatments should have many effects on plant development, I think that the authors need to look at the effect of CK on WUS levels in the WT and in the *da1* mutant in a much shorter timeframe (such as the one shown in Fig.4). They could also show the effect of CK treatments on WUS transcript and protein levels in their DA1 overexpressors.

- Regarding reproducibility, in almost all cases, the number of biological replicates is stated but there is no information on whether each experiment was carried independently several times. Each experiment should be carried at least two independent times and the results of these two replicates should be shown.

- Introduction, the authors talk about HAM/WUS dimers regulating stem cell homeostasis, but they should also mention the recent work from Su and colleagues (<https://www.pnas.org/doi/10.1073/pnas.2015248117>) showing that STM-WUS dimers regulate CLV3 expression and SAM function.

- Fig. 1, The number of SAMs that were imaged in the in situ hybridizations is not displayed. Also, if the authors want to state that more cells are expressing CLV3 in the *da1* mutant, they need to quantify it (which can be performed according to Geier et al, 2008: <https://journals.plos.org/plosone/article?id=10.1371/journal.pone.0003553>)

- Fig. 2c, as *wus-1* mutant does not have any SAM, I am not sure what the cross between *da1-1* and *wus-1* really shows. If DA1 would act independently from WUS in the regulation of SAM size, the double mutant would probably still have no SAM given the strength of *wus-1*. I find the analysis of the cross between *da1-1* and the weak *wus* mutant allele *wus-7* in Fig. S6 more informative (also see comment below).

- Fig. 4, same comment as the one of Fig.1 for the confocal pictures of pWUS::WUS-GFP. There is no mention of the number of SAM that were imaged.

- Fig.4c, shouldn't there be a control for the estradiol treatment (untreated plant or plants not expressing the MYC-DA1 but treated with estradiol)?

- Fig. S2 is showing representative pictures but there is no mention of the number of replicates for SAM and IM size in the da1 mutants.

- Fig. S3, representative pictures would be nice.

- Fig. S6 and text: "The da1-1 Ler wus-7 double mutant showed similar SAM shape phenotypes to the wus-7 single mutant (Figure S6). wus-7 also strongly suppressed the large SAM phenotype of da1-1Ler." I would interpret it differently, da1-1 is partly complementing the wus7 phenotype (da1-1 wus-7 SAM being slightly larger than wus-7 SAM). This actually goes with the function of DA1 described afterwards.

- Material and methods, the way SAM size (in area) was measured is not explained.

Reviewer #2 (Remarks to the Author):

Comments to authors

In this manuscript, Cui et al. investigate the role of the metallopeptidase DA1 in regulating meristem size via WUS degradation. Phenotypic and genetic analysis with different clv and wus mutants showed that DA1 function in the regulation of meristem size in Arabidopsis. WUS and DA1 have a direct physical interaction and DA1 cleaved WUS protein, regulating its homeostasis and decreasing WUS protein levels. The authors then show that expression of DA1 by the WUS promoter leads to smaller meristems, but this does not occur when a mutated version of DA1 is used. Finally, the authors show that cytokinin decreases the level of DA1 but increases its expression.

The finding that a peptidase regulates WUS stability and meristem size regulation is novel and interesting, and it adds an additional layer of complexity to a very well characterized pathway. The manuscript is written in an ok English but in several instances, it needs editing. However, I have several major issues with this manuscript that are highlighted below.

Major concerns:

-The interaction of DA1 and WUS is convincingly shown by different strategies. However, there is no exploration of the domains that drive the physical interaction between the two proteins. This is one shortcoming of this manuscript; WUS has several well characterized domains including a crystal structure of the DNA binding domain, the WUS-box and the EAR-repression motif. This should be explored to better define the structure-function relationship of WUS proteins.

-There is no exploration or determination of where the WUS protein is cleaved. This is another shortcoming.

-Does the 35S::GFP-DA1 line rescue the *da1* phenotype? If this is discussed in the previous DA1 publication from this group, it should be nonetheless mentioned here.

-Figure 5 is not clear to this reader. The pictures are dark, and the red lines prevent an examination of the SAM phenotype.

-In the model in Fig. 6h, the direct connection between cytokinin signaling and WUS expression is removed. Based on the drawn model, it appears that cytokinin signaling regulates WUS1 only via DA1 activity and that is not the case.

-In several instances, the error bars in the control sample (see Fig. 2d, 4b, 4h, 4i, 6c) are non-existent and all data points line up with the value 1. This is not the way the data should be presented. The control sample value is 1 but the variation should be represented as in all the other samples. Similarly, in Fig. 6c the relative expression levels of DA1 in the Col + 6-BA sample do not show any variation; this is hard to believe.

-Fig. 6d: it is not clear how many times the quantification of WUS levels in wt and *da1* 6-BPA treated sample was done.

-How does cytokinin repress the levels of DA1 protein? This is not discussed, even though cytokinin activates the expression of the DA1 gene. How is this may occur is not mentioned.

Minor issues:

-Line 87: correct "HAIRY MERISTEM"

-Lines 182-183: rewrite sentence

-Line 201: explain what DA1_{pep} is when first introduced in the text, not after.

Reviewer #3 (Remarks to the Author):

The manuscript by Cui and colleagues reports on the effect of the DA1 peptidase on shoot meristem development. The authors show data on mRNA expression, SAM phenotypes, genetic interaction with WUSCHEL and CLAVATA mutants, as well as biochemical interaction of DA1 with WUSCHEL and its potential cleavage. In addition, the authors show data on the influence of cytokinin on DA1 protein stability.

While the manuscript contains a number of interesting datasets, the take home message, namely that DA1 controls SAM size via degradation of WUS, is not sufficiently supported so far. The following major issues bring me to this conclusion:

1. The SAM phenotypes shown in Figure 1 don't match the quantification. For example, the *da1-1* SAM shown in panel j is easily twice as large as the wt SAM in panel i, however that SAM area of *da1-1* is only 20% larger than wt if you read the quantification in panel d. It follows that either the quantification is wrong, or the authors do not show representative images to illustrate the phenotypes and expression patterns (for which quantifications are lacking).
2. The genetic interactions shown are not as clear cut as described by the authors. Most importantly, the *da1-1, wus-1* double mutant is clearly distinct from *wus-1* single mutants at the level of SAM organization. This finding argues strongly against the conclusion that the effect of DA1 is mainly through WUS.
3. The observation that WUS targets, such as CLV3, are mis-regulated in *da1-1* is not strong evidence that DA1 acts through WUS, since the expression of these genes scales with meristem size.
4. Similarly, the observation that WUS accumulates to higher levels in *da1-1* mutants could easily be explained by the indirect effect of the larger meristem. The data on pWUS:WUS-GFP distribution is insufficient and needs to be improved with much better microscopy and solid quantification. In addition, the authors need to test the specificity of the WUS antiserum. Given the fact that WUS protein accumulates only to very low amounts in the SAM, it appears surprising that a western blot can pick up the signal so easily.
5. The data on short term response of WUS to inducing DA1 expression suffers from the lack of a mock treatment control, making it impossible to judge the effect of DA1.

6. The data on expression of DA1 from the WUS promoter do not add much, since the authors have already shown that overexpression of DA1 causes smaller SAMs.

7. The data on the interaction with cytokinin signaling are somewhat confusing. It appears that the underlying experiment, namely treating seedlings with BA for 3-7 days in liquid culture does not allow to draw solid conclusions, since a whole plethora of indirect effects will manifest themselves under these conditions. consequently, there are a number of inconsistencies that undermine the conclusions: Firstly, the effect of BA on DA1 protein appears to be highly variable as shown in panel b. Is there any reason? Secondly, the response of WUS to BA at the mRNA and protein level is miles apart. While WUS mRNA is shown to increase by a factor of 7, WUS protein levels increase only by a factor of 2. While this could be biologically relevant, it could also point to a lack of specificity/sensitivity of the antibody and needs to be further investigated using different tools. Thirdly, another glaring inconsistency is that in this experiment, in the *da1-1* mutant WUS protein only increases by 42%, but the SAM is enlarged by 50%, in contrast to the 2.5 fold increase of WUS protein and a corresponding 20% SAM enlargement reported in figures 1 and 4. In addition, BA treatment is shown to increase WUS protein levels by the factor of 2, which supposedly leads to an increase in SAM area of 43%, again not fitting with the WUS to SAM size ratio reported in other experiments of the manuscript.

Dear Reviewers

We would like to thank the reviewers for helpful comments and suggestions. We have carefully considered these suggestions in this revision. We conducted the experiments suggested by the reviewers and addressed your concerns. They have helped make a better paper.

REVIEWER COMMENTS

Reviewer #1 (Remarks to the Author):

The manuscript from Cui and colleagues unravels a new mechanism of regulation of the key meristematic gene WUS which involves DA1, a peptidase previously reported as an organ size regulator. The work of the authors supports that 1. DA1 regulates stem cell activity in the SAM, 2. DA1 physically interacts with WUS and cleaves it, 3. DA1 is involved in post-transcriptional regulation of WUS by CK.

Overall, I find this study very interesting as it not only provides a new mechanism of regulation of WUS but also identifies a new player involved in the post-transcriptional regulation of WUS by CK. However, I wonder if the authors could not go even further to test the functional relevance of DA1-dependent regulation of WUS by CK proposed in their final model, as several articles have showed that WUS levels can be modulated in response to various stresses through CK. The article from Pfeiffer and colleagues (<https://elifesciences.org/articles/17023>) notably shows that light and metabolic signals can influence WUS levels through CK. Have the authors tried to look at the response of WUS to these signals in their lines where DA1 was misregulated? Moreover, the data from Fig.6 also supports that CK regulates DA1 in a post-transcriptional manner. Could the author discuss (or even better provide data), on the mechanisms through which DA1 could be regulated by CK?

Re: Thank you very much for your supportive comments and helpful suggestions. As suggested by the reviewer, we discussed the possible mechanism by which CK regulates DA1 and the possibility that CK-DA1-WUS might mediate environmental signals to regulate stem cell function in this revision. It is also the promising research direction for us to follow this project in the future.

Regarding the experiments presented in the paper, I have issues that would need to be addressed:

- I find most of the westerns blots of Fig. 3 not very clear. In the panel b and c, what are the additional bands we see, notably when WUS and DA1 are together? Are they product of degradation of WUS or non-specific binding? In panel 3d, why is there such a strong band corresponding to GFP alone in the right lane when GFP-DA1 is expressed? Is DA1-GFP cleaved? And what is the intermediate line between the GFP and the DA1-GFP? In panels e and f, we can hardly see the bands. Given the quality of the blots presented (especially Fig.3e and Fig.3f), I think that additional replicates are needed

(notably independent experiments, see further comment on reproducibility). I also think that Nature Communications requires to show the raw blots, that I could not find.

Re: In Fig 3b, c, the additional bands might be the degraded products. The degraded products were normally observed during pull-down or Co-IP experiments in many studies ¹.

In Fig 3d, DA1 is an unstable protein, and GFP might be stable. Thus, the degraded products of DA1-GFP (strong band) might be a GFP, and the intermediate lines between the GFP and the DA1-GFP were GFP with a part of DA1. The similar phenomena were observed in several previous studies ^{2,3}.

In Fig 3e, f, the molecular weight of cleaved product was quite little. We can detect the cleaved product with longer exposure. We further confirmed the WUS cleavage by DA1 using FLAG-WUS and WUS-FLAG (Fig S8). We also showed raw blot in this revision.

- The article provides data supporting that the effect of DA1 on WUS is not transcriptional (especially Fig. 4c-i). However, altering WUS activity will affect SAM size, which should lead to changes in the number of WUS-expressing cells, which will ultimately influence WUS expression and protein levels (in a kind of positive feedback loop). I think that the authors should consider that point when interpreting the results of Fig. 2d (expression of WUS targets in DA1 mutants) and that they should provide data on the expression of WUS in the *dal* mutant together with the data shown in Fig.4 (by qPCR or/and through the analysis of a transcriptional reporter for WUS).

Re: As suggested by the reviewer, we detected expression of *WUS* in this revision (Fig 2e).

- The data from the CK treatments in Fig.6 are quite important for the model presented in Fig.6h. If I understood correctly the material and methods, they were obtained by performing CK treatment for 7 days, which is a rather long time. I understand that few days are needed to see an effect of CK on SAM size (although 7 days is still a lot) but this is not the case for WUS as the data from Fig. 4 supports that DA1 can induce changes in WUS protein levels in less than 8h. Given my previous comment and the fact that long-term CK treatments should have many effects on plant development, I think that the authors need to look at the effect of CK on WUS levels in the WT and in the *dal* mutant in a much shorter timeframe (such as the one shown in Fig.4). They could also show the effect of CK treatments on WUS transcript and protein levels in their DA1 overexpressors.

Re: As suggested by the reviewer, we conducted short term CK treatment (8 hrs) and investigated the numbers of cells with WUS-GFP in the SAMs of *pWUS:WUS-GFP* and *pWUS:WUS-GFP; dal-1* as well as *WUS* transcription in this revision (Fig 6e-6h,

6m-6n). The results supported that DA1 is involved in the regulation of CK-induced WUS protein accumulation. Considering that the reviewer 3 concerns the specificity of WUS antibody, we used *pWUS:WUS-GFP* line to observe cells with WUS-GFP in this revision. Currently, we are transforming *35S:DA1* to *pWUS:WUS-GFP*. It will take times to get *35S:DA1; pWUS:WUS-GFP* homozygous plants. Because of the revision time requirement, we hope that the reviewer could understand this situation.

- Regarding reproducibility, in almost all cases, the number of biological replicates is stated but there is no information on whether each experiment was carried independently several times. Each experiment should be carried at least two independent times and the results of these two replicates should be shown.

Re: As suggested by the reviewers, the results of replicates were described in figure legends and shown in Supplemental figures in this revision.

- Introduction, the authors talk about HAM/WUS dimers regulating stem cell homeostasis, but they should also mention the recent work from Su and colleagues (<https://www.pnas.org/doi/10.1073/pnas.2015248117>) showing that STM-WUS dimers regulate CLV3 expression and SAM function.

Re: We cited this paper in this revision.

- Fig. 1, The number of SAMs that were imaged in the in situ hybridizations is not displayed. Also, if the authors want to state that more cells are expressing CLV3 in the *da1* mutant, they need to quantify it (which can be performed according to Geier et al, 2008: <https://journals.plos.org/plosone/article?id=10.1371/journal.pone.0003553>)

Re: As suggested by the reviewer, we quantified the number of fluorescing cells in the wild type and *da1-1* using *pCLV3:YFP* marker line. In this revision, we observed expression of *pDA1:GUS* and found that DA1 has strong expression in SAM in independent lines (Fig 1c-1e).

- Fig. 2c, as *wus-1* mutant does not have any SAM, I am not sure what the cross between *da1-1* and *wus-1* really shows. If DA1 would act independently from WUS in the regulation of SAM size, the double mutant would probably still have no SAM given the strength of *wus-1*. I find the analysis of the cross between *da1-1* and the weak *wus* mutant allele *wus-7* in Fig. S6 more informative (also see comment below).

Re: As suggested by the reviewer, we showed the genetic data for *da1-1* and weak allele *wus-7* in main Figure 2c, d and moved the genetic data for the *da1-1* and a strong allele *wus-1* to Supplemental Figure 5.

- Fig. 4, same comment as the one of Fig.1 for the confocal pictures of *pWUS::WUS-*

GFP. There is no mention of the number of SAM that were imaged.

Re: As suggested by the reviewer, we quantified the number of fluorescing cells in the wild type and *dal-1* SAM and IM using *pWUS:WUS-GFP* line and showed the number of SAM in Figure legends.

- Fig.4c, shouldn't there be a control for the estradiol treatment (untreated plant or plants not expressing the MYC-DA1 but treated with estradiol)?

Re: We further quantified the number of fluorescing cells in the SAMs of *pWUS:WUS-GFP* and estradiol inducible *MYC-DA1; pWUS:WUS-GFP* line with or without estradiol treatment. When treated with estradiol, the number of fluorescing cells in the SAMs of *MYC-DA1; pWUS:WUS-GFP* was dramatically decreased (Fig 4g-h, 4l). By contrast, the estradiol treatment did not influence the number of fluorescing cells in the SAMs of *pWUS:WUS-GFP* (Figure 4i-j, 4m).

- Fig. S2 is showing representative pictures but there is no mention of the number of replicates for SAM and IM size in the *dal* mutants.

Re: We added the number of replicates for SAM and IM size in this revision (Fig S2).

- Fig. S3, representative pictures would be nice.

Re: As suggested by the reviewer, we added representative pictures in this revision (Fig 1a).

- Fig. S6 and text: "The *dal-1 Ler wus-7* double mutant showed similar SAM shape phenotypes to the *wus-7* single mutant (Figure S6). *wus-7* also strongly suppressed the large SAM phenotype of *dal-1Ler*." I would interpret it differently, *dal-1* is partly complementing the *wus7* phenotype (*dal-1 wus-7* SAM being slightly larger than *wus-7* SAM). This actually goes with the function of DA1 described afterwards.

Re: As shown in Fig 2d, the SAM size in *dal-1^{Ler}* was increased by about 60% compared to *Ler*, while the SAM size in *dal-1^{Ler} wus-7* was increased by about 40% compared with *wus-7*, indicating that DA1 and WUS have overlapped function in SAM size control. The *dal-1^{Ler} wus-7* had similar distorted SAM phenotype of *wus-7* (Figure 2c). In addition, we also showed that DA1 can cleave and destabilize WUS, suggesting that WUS may act downstream of DA1. In this revision, we claimed that these genetic analyses indicated that the large SAM of *dal-1* partially depends on the functional *WUS*.

- Material and methods, the way SAM size (in area) was measured is not explained.

Re: As suggested by the reviewer1, we described the detailed method for SAM measurement and showed the measured region of SAM in this revision (Fig S3).

Reviewer #2 (Remarks to the Author):

Comments to authors

In this manuscript, Cui et al. investigate the role of the metallopeptidase DA1 in regulating meristem size via WUS degradation. Phenotypic and genetic analysis with different *clv* and *wus* mutants showed that DA1 function in the regulation of meristem size in Arabidopsis. WUS and DA1 have a direct physical interaction and DA1 cleaved WUS protein, regulating its homeostasis and decreasing WUS protein levels. The authors then show that expression of DA1 by the WUS promoter leads to smaller meristems, but this does not occur when a mutated version of DA1 is used. Finally, the authors show that cytokinin decreases the level of DA1 but increases its expression.

The finding that a peptidase regulates WUS stability and meristem size regulation is novel and interesting, and it adds an additional layer of complexity to a very well characterized pathway. The manuscript is written in an ok English but in several instances, it needs editing. However, I have several major issues with this manuscript that are highlighted below.

Re: Thanks for your supportive and helpful comments. We also carefully edited the English in this revision.

Major concerns:

-The interaction of DA1 and WUS is convincingly shown by different strategies. However, there is no exploration of the domains that drive the physical interaction between the two proteins. This is one shortcoming of this manuscript; WUS has several well characterized domains including a crystal structure of the DNA binding domain, the WUS-box and the EAR-repression motif. This should be explored to better define the structure-function relationship of WUS proteins.

Re: Thanks for your helpful suggestions. As suggested by the reviewer, we performed Y2H assay to identify the interacting domains (Fig S6). Considering that WUS contains the homeodomain, the dimerization domain, the WUS-box and the EARL motif, we asked which domain of WUS could interact with DA1. Interestingly, we found that the C-terminal region of DA1 (LIM domains and the peptidase domain) interacted with the region with both the dimerization domain and the C terminal region (WUS-box and EARL motif), although the C-terminal region of DA1 did not interact with any single domain of WUS (Figure S6). Similar phenomena have been reported in previous studies⁴.

-There is no exploration or determination of where the WUS protein is cleaved. This is another shortcoming.

Re: Thanks for your suggestions. We tried best to identify the cleavage site of the WUS protein. Considering that we only got every little amount of cleaved products, it failed

to identify the cleavage site. We hope that the reviewer could understand this situation.

-Does the 35S::GFP-DA1 line rescue the *da1* phenotype? If this is discussed in the previous DA1 publication from this group, it should be nonetheless mentioned here.

Re: Thanks for your suggestions. A study showed that *35S::GFP-DA1* formed small organs^{2,5}, indicating GFP-DA1 is functional. As suggested by the reviewer, we cited these papers.

-Figure 5 is not clear to this reader. The pictures are dark, and the red lines prevent an examination of the SAM phenotype.

Re: Thanks for your suggestions. As suggested by the reviewer, we removed red lines.

-In the model in Fig 6h, the direct connection between cytokinin signaling and WUS expression is removed. Based on the drawn model, it appears that cytokinin signaling regulates WUS1 only via DA1 activity and that is not the case.

Re: Thanks for your suggestions. We redraw the model. Cytokinin signaling can regulate WUS stability by DA1 and also regulate the transcription of the *WUS* gene (Figure 6p).

-In several instances, the error bars in the control sample (see Fig. 2d, 4b, 4h, 4i, 6c) are non-existent and all data points line up with the value 1. This is not the way the data should be presented. The control sample value is 1 but the variation should be represented as in all the other samples. Similarly, in Fig. 6c the relative expression levels of DA1 in the Col + 6-BA sample do not show any variation; this is hard to believe.

Re: Thanks for your suggestions. We edited these and showed variation in this revision.

-Fig. 6d: it is not clear how many times the quantification of WUS levels in wt and *da1* 6-BPA treated sample was done.

Re: In this revision, according to the reviewer's suggestions, we investigated the number of fluorescing cells of SAMs in *pWUS::WUS-GFP* (n=14), *pWUS::WUS-GFP* + 6-BA (n=13), *pWUS::WUS-GFP; da1-1* (n=9), and *pWUS::WUS-GFP; da1-1* + 6-BA (n=13) transgenic plants with or without 6-BA treatments (Figure 6e-6h, 6m). We got similar conclusions.

-How does cytokinin repress the levels of DA1 protein? This is not discussed, even though cytokinin activates the expression of the DA1 gene. How is this may occur is not mentioned.

Re: Thanks. We discussed the possible mechanism by which cytokinin signaling

regulates the levels of DA1 in the discussion in this revision.

Minor issues:

-Line 87: correct "HAIRY MERISTEM"

Re: Thanks for spotting this. We edited this.

-Lines 182-183: rewrite sentence

Re: Thanks. We rewrote this sentence.

-Line 201: explain what DA1pep is when first introduced in the text, not after.

Re: Thanks. We edited this.

Reviewer #3 (Remarks to the Author):

The manuscript by Cui and colleagues reports on the effect of the DA1 peptidase on shoot meristem development. The authors show data on mRNA expression, SAM phenotypes, genetic interaction with WUSCHEL and CLAVATA mutants, as well as biochemical interaction of DA1 with WUSCHEL and its potential cleavage. In addition, the authors show data on the influence of cytokinin on DA1 protein stability.

While the manuscript contains a number of interesting datasets, the take home message, namely that DA1 controls SAM size via degradation of WUS, is not sufficiently supported so far. The following major issues bring me to this conclusion:

1. The SAM phenotypes shown in Figure 1 don't match the quantification. For example, the *dal-1* SAM shown in panel j is easily twice as large as the wt SAM in panel I, however that SAM area of *dal-1* is only 20% larger than wt if you read the quantification in panel d. It follows that either the quantification is wrong, or the authors do not show representative images to illustrate the phenotypes and expression patterns (for which quantifications are lacking).

Re: Thank you for spotting this. We carefully checked raw data and found that the images came for the seedlings with different growth ages or time. This promoted us to investigate the differences between wild-type and *dal-1* SAM size in 6-day-old and 9-day-old seedlings. We found that the largest differences between wild-type and *dal-1* SAM size were observed in 6-d old seedlings. Thus, we used 6-d-old seedlings to measure the SAM size in this revision.

The *da1-1* forms large shoot apical meristem. **a** Morphologies of SAMs of Col-0 and *da1-1*. Plants were grown for 6 days or 9 days in long-day conditions. **b** Morphologies of SAMs of Ler and *da1-1^{Ler}*. Plants were grown for 6 days and 9 days in long-day conditions. **c** The average area of Col-0 and *da1-1* SAMs (n = 38). Plants were grown for 6 days in long-day conditions. **d** The average area of Col-0 and *da1-1* SAMs (n = 38). Plants were grown for 9 days in long-day conditions. **e** The average area of Ler and *da1-1^{Ler}* SAMs (n = 43). Plants were grown for 6 days in long-day conditions. **f** The average area of Ler and *da1-1^{Ler}* SAMs (n = 55). Plants were grown for 9 days in long-day conditions.

2. The genetic interactions shown are not as clear cut as described by the authors. Most importantly, the *da1-1*, *wus-1* double mutant is clearly distinct from *wus-1* single mutants at the level of SAM organization. This finding argues strongly against the conclusion that the effect of DA1 is mainly through WUS.

Re: As suggested by other reviewers, the *wus-1* allele is very strong allele and has no typical SAM, and it is better to use the weak allele *wus-7* to conduct the genetic analyses. As shown in figure 2d, the SAM size in *da1-1^{Ler}* was increased by about 60% compared to Ler, while the SAM size in *da1-1^{Ler} wus-7* was increased by about 40% compared with *wus-7*, supporting that DA1 and WUS have partially overlapped function in SAM size control. Similar genetic analyses have been described previously⁶. The *da1-1^{Ler} wus-7* had similar distorted SAM phenotype of *wus-7* (Figure 2c). In addition, we also

showed that DA1 can cleave and destabilize WUS, suggesting that WUS may act downstream of DA1.

3. The observation that WUS targets, such as CLV3, are mis-regulated in *da1-1* is not strong evidence that DA1 acts through WUS, since the expression of these genes scales with meristem size.

Re: The expression region of *CLV3* represents stem cell populations. Consistent with your suggestions, we used *CLV3* as a marker gene to explain that DA1 influences stem cell number or population. As shown in Fig 1e, the number of cells with *CLV3* expression in *da1-1* is increased, indicating that *da1-1* has more stem cells in SAM than the wild type.

4. Similarly, the observation that WUS accumulates to higher levels in *da1-1* mutants could easily be explained by the indirect effect of the larger meristem. The data on pWUS:WUS-GFP distribution is insufficient and needs to be improved with much better microscopy and solid quantification. In addition, the authors need to test the specificity of the WUS antiserum. Given the fact that WUS protein accumulates only to very low amounts in the SAM, it appears surprising that a western blot can pick up the signal so easily.

Re: We agree with this. The large SAM of *da1-1* may contain more cells with WUS expression, which might cause the increased accumulation of WUS proteins. To address this question, we generated transgenic lines of the estradiol-inducible form of DA1 (*pER8:MYC-DA1*) and then crossed with *pWUS:WUS-GFP* to obtain *pWUS:WUS-GFP; pER8:MYC-DA1* plants. The *pWUS:WUS-GFP; pER8:MYC-DA1* seedlings treated with β -estradiol treatment for 12 h led to the strong accumulation of MYC-DA1 protein (Figure 4k). Conversely, cells with WUS-GFP in the SAMs of *pWUS:WUS-GFP; pER8:MYC-DA1* were dramatically decreased at 12 h after induction (Figure 4g, 4h, 4l), although the level of *WUS* mRNA was not reduced after induction (Figure 4n). By contrast, cells with WUS-GFP in the SAMs of *pWUS:WUS-GFP* (a negative control) was not changed at 12 h after β -estradiol treatment (Figure 4i, 4j, 4m). Considering that the β -estradiol treatment for 12 h did not obviously affect SAM size (Figure 4g-4j), the β -estradiol induced decrease in cells with WUS-GFP in the SAMs of *pWUS:WUS-GFP; pER8:MYC-DA1* was not caused by SAM size. Thus, DA1 activity destabilizes WUS proteins in Arabidopsis. Considering that the reviewer3 concerns the specificity of the WUS antiserum, we used *pWUS:WUS-GFP* to monitor WUS protein levels as described in previous studies in this revision ⁷.

5. The data on short term response of WUS to inducing DA1 expression suffers from the lack of a mock treatment control, making it impossible to judge the effect of DA1.

Re: As suggested by the reviewer3, we quantified cells with WUS-GFP in the SAMs of estradiol inducible *MYC-DA1; pWUS:WUS-GFP* line and *pWUS:WUS-GFP* (control)

with or without short-time (8h) estradiol treatment in this revision (Fig 4g-o). We got similar conclusions.

6. The data on expression of DA1 from the WUS promoter do not add much, since the authors have already shown that overexpression of DA1 causes smaller SAMs.

Re: Considering that *DA1* is expressed in the whole meristem, we asked whether expression of *DA1* in the *WUS* expression domain could influence *WUS* function and SAM size. Consistent with this notion, the expression of *DA1* driven by the *WUS* promoter can complement the SAM phenotype of *dal-1*, indicating that expression of *DA1* in the *WUS* expression domain influences *WUS* function and SAM size and also supporting that DA1 and WUS have intimate relationship.

7. The data on the interaction with cytokinin signaling are somewhat confusing. It appears that the underlying experiment, namely treating seedlings with BA for 3-7 days in liquid culture does not allow to draw solid conclusions, since a whole plethora of indirect effects will manifest themselves under these conditions. consequently, there are a number of inconsistencies that undermine the conclusions: Firstly, the effect of BA on DA1 protein appears to be highly variable as shown in panel b. Is there any reason?

Re: The triplicate repeats were performed by two students (Yu Li and Gucai Cui), respectively. They used seedlings with different growth stages to detect DA1 proteins, respectively. Nonetheless, they got similar conclusion that CK represses the level of DA1. In this revision, we repeated this experiments using biological replicates with similar growth conditions (Fig 6a, S9).

Secondly, the response of WUS to BA at the mRNA and protein level is miles apart. While WUS mRNA is shown to increase by a factor of 7, WUS protein levels increase only by a factor of 2. While this could be biologically relevant, it could also point to a lack of specificity/sensitivity of the antibody and needs to be further investigated using different tools.

Re: Considering that the reviewer3 concerns the specificity of WUS antibody, we used *pWUS::WUS-GFP* and *pWUS::WUS-GFP; dal-1* to monitor WUS protein in this revision (Fig4a, b) ⁷. When treated with 6-BA for 8 h, cells with WUS-GFP in the SAMs of *pWUS::WUS-GFP* was strongly increased compared with untreated control, whereas cells with WUS-GFP in the SAMs of *pWUS::WUS-GFP; dal-1* was only slightly increased (Figure 6e-6h, 6m). By contrast, the transcripts of *WUS-GFP* in *pWUS::WUS-GFP* and *pWUS::WUS-GFP; dal-1* plants were induced by 6-BA (8h) in a similar manner (Figure 6n). Considering that WUS protein level could be influenced by the regulation of transcriptional level and post-transcriptional level, it seems plausible that WUS protein level did not show the same proportion change as *WUS* mRNA level.

Thirdly, another glaring inconsistency is that in this experiment, in the *dal-1* mutant

WUS protein only increases by 42%, but the SAM is enlarged by 50%, in contrast to the 2.5 fold increase of WUS protein and a corresponding 20% SAM enlargement reported in figures 1 and 4. In addition, BA treatment is shown to increase WUS protein levels by the factor of 2, which supposedly leads to an increase in SAM area of 43%, again not fitting with the WUS to SAM size ratio reported in other experiments of the manuscript.

Re: Thank you for spotting this. As mentioned above, the SAM images came from the seedlings with different growth ages or time in different figures, because the student Guicai Cui continues to conduct these experiments after Yu Li graduated. They used different growth conditions and growth time. Nonetheless, they got similar conclusion that DA1 controls SAM size by influencing WUS protein level. In this revision, we repeated all the related experiments under similar growth conditions and growth time. We also used *pWUS:WUS-GFP* and *pWUS::WUS-GFP; da1-1* lines to monitor WUS protein as described previously ⁷. In addition, it is plausible that WUS-GFP levels in mutants or under 6-BA treatments might not have exactly same proportion effect on SAM size, because the SAM size could be influenced by growth stages, growth time and growth conditions.

References

- 1 Jia, T., Li, F., Liu, S., Dou, J. & Huang, T. DnaJ Proteins Regulate WUS Expression in Shoot Apical Meristem of Arabidopsis. *Plants (Basel)* **10**, doi:10.3390/plants10010136 (2021).
- 2 Dong, H. *et al.* Ubiquitylation activates a peptidase that promotes cleavage and destabilization of its activating E3 ligases and diverse growth regulatory proteins to limit cell proliferation in Arabidopsis. *Genes Dev* **31**, 197-208, doi:10.1101/gad.292235.116 (2017).
- 3 Du, L. *et al.* The ubiquitin receptor DA1 regulates seed and organ size by modulating the stability of the ubiquitin-specific protease UBP15/SOD2 in Arabidopsis. *Plant Cell* **26**, 665-677, doi:10.1105/tpc.114.122663 (2014).
- 4 Zhou, Y. *et al.* Control of plant stem cell function by conserved interacting transcriptional regulators. *Nature* **517**, 377-380, doi:10.1038/nature13853 (2015).
- 5 Vanhaeren, H. *et al.* UBP12 and UBP13 negatively regulate the activity of the ubiquitin-dependent peptidases DA1, DAR1 and

DAR2. *Elife* **9**, doi:10.7554/eLife.52276 (2020).

- 6 Lin, T. F., Saiga, S., Abe, M. & Laux, T. OBE3 and WUS Interaction in Shoot Meristem Stem Cell Regulation. *PLoS One* **11**, e0155657, doi:10.1371/journal.pone.0155657 (2016).
- 7 Daum, G., Medzihradzky, A., Suzaki, T. & Lohmann, J. U. A mechanistic framework for noncell autonomous stem cell induction in Arabidopsis. *Proc Natl Acad Sci U S A* **111**, 14619-14624, doi:10.1073/pnas.1406446111 (2014).

REVIEWER COMMENTS

Reviewer #1 (Remarks to the Author):

In this revised version of their manuscript, Cui and colleagues have answered to almost all of my concerns by providing new experiments that really strengthen their conclusion. The only major concern that I have is still regarding reproducibility. The authors have displayed the number of biological replicates that they have in each experiment, have performed quantifications when needed and do statistical tests. However, it is still not clear to me if each experiment was carried twice on two independent batches of plants, which, from my point of view, is a prerequisite for publication. Apart from this major point, I only have minor comments to help clarifying specific points in the text:

-The manuscript would really profit from a correction of its English as there are many mistakes of grammar.

-Regarding the expression of DA1, which is observed by GUS staining, it is hard to see something (we can just see that everything is blue). I think the author should show us a SAM not expressing any construct (or even better expressing a GUS under the control of a promoter which is not expressed in the SAM) as a control so that we can really see what is the GUS signal on the figure

-I do not fully agree with the interpretation of da1-1 wus-7 phenotype. da1-1 can very partly complement the phenotype of the weak wus-7 as da1-1 wus7 has statistically larger SAMs than wus7. This makes sense as wus-7 is not a null allele. The remaining activity of WUS in this weak mutant allele should still be influenced by da1-1 (explaining the small increase in meristem size in the double mutant compared to wus-7)

-It is nice that the authors explained the presence of additional bands of WUS-GFP with some references in their response to reviewers but it would be nice if they also discuss this in the main text with a sentence or two (as readers may be asking the same question).

-I thank the authors for this additional replicate showing the cleavage of WUS by DA1 (Fig.S8), I would suggest that they invert which replicate is in main and which replicate is in Supp (as the one in Supp is more convincing)

-Fig.6b has a mistake, it is written "Relative WUS protein levels" where it should be "Relative DA1 protein levels"

-In Fig 6, as different times of treatments are used for SAM size and for WUS-GFP level measurements (3 days vs 8h), I would encourage the authors to put these times on the figure panels so that the readers do not get confused (as time really matters here to show that DA1 directly modulate WUS levels).

-End of the results and discussion: I think it is important to not minimize the induction of WUS expression by CK which is also clearly seen here. I would thus amend the last sentence of the results to really state that CK can promote SAM enlargement by inducing the expression of WUS and by increasing the stability of the WUS protein through the repression of DA1 activity.

-Discussion: Line 346: “CK signalling promotes the accumulation of WUS protein, at least in part, through the repression of DA1 protein” The authors should expand a bit the: “at least in part” as they really cannot rule out the possibility that CK could also affect WUS stability through another mechanism.

Reviewer #2 (Remarks to the Author):

The authors adequately addressed most of my previous concerns regarding this manuscript. Here are a few items that should be addressed:

- Fig.1 c is not informative, as shown: it is difficult to see what the reader is looking at and the GUS staining is seen everywhere. The authors should provide more GUS staining pictures of whole seedlings to show whether DA1 expression is specific to meristems and young leaf primordia or is expressed in every organ. This should be provided as a Supplemental Figure.
- It is unclear to me how the quantification of WUS expressing cells was performed. Was it done with meristem confocal sections? Or in 3D rendering of SAMs? Please provide details on the quantification.
- Please provide in Supplemental Fig. 6 the map of the DA1 protein domain together with WUS.

Minor issues:

- line 74: correct “influence”

Reviewer #3 (Remarks to the Author):

This version of the manuscript by Cui and colleagues is substantially improved and now makes a good case for the regulation of global WUSCHEL protein levels by the DA1 peptidase. However, it is unfortunately still far from publication ready, since a number of critical flaws remain.

1. Fig1 c: pDA1:GUS staining is not visible and much better pictures are required.

2. The genetic interaction of WUS and DA1 is presented in a very oversimplified manner. Genetic interactions of weak alleles are notoriously difficult to interpret. More important are studies with clean loss of function alleles, as described in the initial version. Unfortunately, the authors now hide this data in the supplements and have reduced the quality of the analysis. What happened to the high-quality meristem sections of *wus-1* and *da1/wus1*? They are now replaced by much worse pictures, that do not allow us to see the meristem structure, just the outline. This is a key piece of information, which needs to be shown and carefully interpreted. All data points to the fact that DA1 can cleave WUS and that some effects on meristem size are due to this interaction, but that other pathways act in parallel (*wus-1/da1* double mutants do not look like *wus1*!) This is not a surprise, since the authors have shown that DA1 also cleaves other proteins, including WOXes. Insisting that all effects are due to cleavage of WUS is not only wrong, but it will not make the manuscript better, or more important.

3. The yeast data aimed at mapping the interaction domains of WUS and DA1 are confusing. Figure 3 supposedly shows that full length WUS interacts with full length DA1, but in FigS6 only a truncated version of DA1 is tested against truncated versions of WUS. Why aren't the full-length versions of one protein tested against the deletion versions of the other? Also, the naming and labelling strategy could be improved.

4. The data on cytokinin influencing DA1 in turn WUS accumulation is not solid. Cytokinin treatment does not dramatically reduce 35S:DA1-GFP accumulation as claimed, but the effects are rather mild and quantification with normalization to the loading control is missing. What is more disturbing is that endogenous DA1 mRNA expression is enhanced more than two-fold, making it impossible to judge how the endogenous DA1 protein levels respond to cytokinin treatment. The conclusion that there will be less DA1 protein due to cytokinin signaling is not supported at all. This fits well with the results on WUS-GFP after cytokinin treatment. Since the wild-type meristems shown are bigger after 8h of cytokinin treatment (which they should not be), there are more WUS positive cells. This difference does not exist in the *da1* mutant, therefore there is no change in WUS-GFP expression. It appears that the experiment was not carried out with enough care to exclude effects from variations in meristem size. The experiments using longer treatment times are not very helpful here, because of the meristem expansion and indirect effects.

Dear Reviewers

We would like to thank the reviewers for helpful comments and suggestions. We have carefully considered these suggestions in this revision. We conducted the experiments suggested by the reviewers and addressed your concerns. They have helped make a better paper.

Reviewer #1 (Remarks to the Author):

In this revised version of their manuscript, Cui and colleagues have answered to almost all of my concerns by providing new experiments that really strengthen their conclusion. The only major concern that I have is still regarding reproducibility. The authors have displayed the number of biological replicates that they have in each experiment, have performed quantifications when needed and do statistical tests. However, it is still not clear to me if each experiment was carried twice on two independent batches of plants, which, from my point of view, is a prerequisite for publication. Apart from this major point, I only have minor comments to help clarifying specific points in the text:

Re: Thanks for your very positive comments. We carefully checked our raw data, and were confident to make a conclusion that each experiment was carried at least twice with independent batches of plants. We also described these in the figures legends as indicated.

-The manuscript would really profit from a correction of its English as there are many mistakes of grammar.

Re: Thanks. We have carefully polished the English language in this revision.

-Regarding the expression of DA1, which is observed by GUS staining, it is hard to see something (we can just see that everything is blue). I think the author should show us a SAM not expressing any construct (or even better expressing a GUS under the control of a promoter which is not expressed in the SAM) as a control so that we can really see what is the GUS signal on the figure

Re: Thanks. As suggested, we sectioned the SAMs of *pDA1:GUS* and the wild type (Col-0, a negative control), and these made them for better visibility (See Fig. 1c).

-I do not fully agree with the interpretation of *da1-1 wus-7* phenotype. *da1-1* can very partly complement the phenotype of the weak *wus-7* as *da1-1 wus7* has statistically larger SAMs than *wus7*. This makes sense as *wus-7* is not a null allele. The remaining activity of WUS in this weak mutant allele should still be influenced by *da1-1* (explaining the small increase in meristem size in the double mutant compared to *wus-7*)

Re: Thanks for your suggestion. We agree with this. As suggested, we explained the small increase in meristem size in the double mutant compared to *wus-7* in this version.

-It is nice that the authors explained the presence of additional bands of WUS-GFP with some references in their response to reviewers but it would be nice if they also discuss this in the main text with a sentence or two (as readers may be asking the same question).

Re: Thanks. We added the explanation about the additional bands of WUS-GFP in the main text.

-I thank the authors for this additional replicate showing the cleavage of WUS by

DA1 (Fig.S8), I would suggest that they invert which replicate is in main and which replicate is in Supp (as the one in Supp is more convincing)

Re: Thanks. As suggested, we exchanged Fig. 3e and 3F with Fig. S8a and S8b (this version Fig. S9) in this revision.

-Fig.6b has a mistake, it is written "Relative WUS protein levels" where it should be "Relative DA1 protein levels"

Re: Thanks for spotting this mistake. We corrected this in this revision.

-In Fig 6, as different times of treatments are used for SAM size and for WUS-GFP level measurements (3 days vs 8h), I would encourage the authors to put these times on the figure panels so that the readers do not get confused (as time really matters here to show that DA1 directly modulate WUS levels).

Re: Thanks. As suggested, we labeled the figure panels with the treated times of CK.

-End of the results and discussion: I think it is important to not minimize the induction of WUS expression by CK which is also clearly seen here. I would thus amend the last sentence of the results to really state that CK can promote SAM enlargement by inducing the expression of WUS and by increasing the stability of the WUS protein through the repression of DA1 activity.

Re: Thanks. We agreed with this, and amended the last sentence as suggested.

-Discussion: Line 346: "CK signalling promotes the accumulation of WUS protein, at least in part, through the repression of DA1 protein" The authors should expand a bit the: "at least in part" as they really cannot rule out the possibility that CK could also affect WUS stability through another mechanism.

Re: Thanks. We agreed with this, and changed the sentence as below:

“...However, we cannot rule out the possibility that CK could also affect WUS stability through another unknown mechanism as CK treatment can still induce the accumulation of the WUS protein and the enlargement of the SAM in *dal-1*, albeit the ratio of induction in *dal-1* was less than that in Col-0 (Figure 6)”.

Reviewer #2 (Remarks to the Author):

The authors adequately addressed most of my previous concerns regarding this manuscript. Here are a few items that should be addressed:

Re: Thanks for your very positive comments.

- Fig.1 c is not informative, as shown: it is difficult to see what the reader is looking at and the GUS staining is seen everywhere. The authors should provide more GUS staining pictures of whole seedlings to show whether DA1 expression is specific to meristems and young leaf primordia or is expressed in every organ. This should be provided as a Supplemental Figure.

Re: Thanks for your suggestions. As suggested, we sectioned the SAMs of *pDA1:GUS* and the wild type (Col-0, a negative control), and these made them for better visibility (See Fig. 1c). We also provided GUS staining pictures of the whole seedlings to show *DA1* expression in young leaves (Supplemental Figure S4).

- It is unclear to me how the quantification of WUS expressing cells was performed. Was it done with meristem confocal sections? Or in 3D rendering of SAMs? Please provide details on the quantification.

Re: Thanks. The quantification of WUS expressing cells was performed with meristem confocal sections, and the details can be found in Methods “Confocal microscope observation”. ?

- Please provide in Supplemental Fig. 6 the map of the DA1 protein domain together with WUS.

Re: Thanks. As suggested, we added the protein domain map of DA1 and WUS in Supplemental Fig. 7 in this revision.

Minor issues:

-line 74: correct “influence”

Re: Thanks. We corrected this.

Reviewer #3 (Remarks to the Author):

This version of the manuscript by Cui and colleagues is substantially improved and now makes a good case for the regulation of global WUSCHEL protein levels by the DA1 peptidase. However, it is unfortunately still far from publication ready, since a number of critical flaws remain.

Re: Thanks for your positive comments.

1. Fig1 c: pDA1:GUS staining is not visible and much better pictures are required.

Re: Thanks. As suggested, we sectioned the SAMs of *pDA1:GUS* and the wild type (Col-0, a negative control), and these made them for better visibility (See Fig. 1c).

2. The genetic interaction of WUS and DA1 is presented in a very oversimplified manner. Genetic interactions of weak alleles are notoriously difficult to interpret. More important are studies with clean loss of function alleles, as described in the initial version. Unfortunately, the authors now hide this data in the supplements and have reduced the quality of the analysis. What happened to the high-quality meristem

sections of *wus-1* and *dal/wus1*? They are now replaced by much worse pictures, that do not allow us to see the meristem structure, just the outline. This is a key piece of information, which needs to be shown and carefully interpreted. All data points to the fact that DA1 can cleave WUS and that some effects on meristem size are due to this interaction, but that other pathways act in parallel (*wus-1/dal* double mutants do not look like *wus1*!) This is not a surprise, since the authors have shown that DA1 also cleaves other proteins, including WOXes. Insisting that all effects are due to cleavage of WUS is not only wrong, but it will not make the manuscript better, or more important.

Re: Thanks for your good suggestions. We agreed that the genetic interaction between *DA1* and *WUS* is a key piece of information for interpreting *DA1* and *WUS* function. Based on your suggestions and Reviewer #1 suggestions in previous version, we made some changes in this revision:

1. In order to easily compare the phenotypes between *dal-1* and strong or weak *wus* mutants, we presented the genetic data of *dal-1* with *wus-1* and *dal-1* with *wus-7* in Figure 2 in this revision.
2. In order to observe the meristem structure, we sectioned the SAM of 6-day-old Ler, *wus-1*, *dal-1^{Ler}* and double mutant. In original version, we sectioned 9-day seedlings, while we used 6-d-old seedlings to observe the SAM size in the revision1. To maintain consistency with meristem size analyses, we re-sectioned 6-d-old seedlings and removed the 9-d section pictures in this revision.
3. We agreed with the reviewer's opinion. DA1 partially depends on the functional WUS and made some changes in the main text part in this revision.

3. The yeast data aimed at mapping the interaction domains of WUS and DA1 are confusing. Figure 3 supposedly shows that full length WUS interacts with full length DA1, but in FigS6 only a truncated version of DA1 is tested against truncated versions of WUS. Why aren't the full-length versions of one protein tested against the

deletion versions of the other? Also, the naming and labelling strategy could be improved.

Re: Thanks. As suggested, we added the yeast interaction data of the full length WUS with the different deletion versions of DA1. However, the full length DA1 fused to BD autoactivates the reporter gene as reported previously ¹. In addition, our results showed that most of deletion versions of WUS fused to BD also autoactivate the reporter gene. Thus, we used DA1-LIM+C fused to BD to test its interactions with WUS, respectively. We also improved the protein domain maps of DA1 and WUS in Supplemental Figure 7 to make better view.

4. The data on cytokinin influencing DA1 in in turn WUS accumulation is not solid. Cytokinin treatment does not dramatically reduce 35S:DA1-GFP accumulation as claimed, but the effects are rather mild and quantification with normalization to the loading control is missing. What is more disturbing is that endogenous DA1 mRNA expression is enhanced more than two-fold, making it impossible to judge how the endogenous DA1 protein levels respond to cytokinin treatment. The conclusion that there will be less DA1 protein due to cytokinin signaling is not supported at all. This fits well with the results on WUS-GFP after cytokinin treatment. Since the wild-type meristems shown are bigger after 8h of cytokinin treatment (which they should not be), there are more WUS positive cells. This difference does not exist in the *da1* mutant, therefore there is no change in WUS-GFP expression. It appears that the experiment was not carried out with enough care to exclude effects from variations in meristem size. The experiments using longer treatment times are not very helpful here, because of the meristem expansion and indirect effects.

Re: Thank you very much for your constructive suggestions. We agreed that the CK longer treatment times are not helpful because 3-day-CK treatment dramatically increased meristem size. The meristem size differences in 3-day CK treated-seedlings may influence DA1 protein and *DAI* gene levels. In this revision, we detected the

levels of DA1 protein and *DA1* gene using samples treated with CK for 8h. Because 8h-CK treatment did not influence SAM size, this excluded the effect of SAM size on DA1 protein and *DA1* gene expression levels. Our results listed as below:

1. The quantification showed that the wild-type meristems had similar the average size with or without CK treatment for 8 h (Supplemental Figure 10). This excluded the effect of SAM size on DA1 protein and *DA1* gene expression levels. In fact, the meristem size had some variations, the pictures we selected did not represent the quantification results in the previous version, and this caused misleading in previous version. We selected representative pictures in this revision.
2. For endogenous *DA1* mRNA expression level, our result showed that *DA1* had similar expression levels with or without CK treatment for 8 h (Figure 6d). By contrast, the levels of DA1 protein were significantly reduced by 8 h-CK treatment (Figure 6a, b). Because 8 h-CK treatment did not influence SAM size, this excluded the effect of SAM size on DA1 protein and *DA1* gene expression levels. Thus, this result supported that CK decreases the level of DA1 protein. In previous version, the alteration of the *DA1* gene expression level after 3- day- CK treatment may be caused by the meristem enlargement, consistent with the reviewer' idea. We used the results 8 h CK treatment in this revision (Figure 6d).
3. As suggested by the reviewer, we removed the results of 3-day- cytokinin-induced accumulation of WUS protein in this revision.

- 1 Peng, Y. *et al.* The ubiquitin receptors DA1, DAR1, and DAR2 redundantly regulate endoreduplication by modulating the stability of TCP14/15 in Arabidopsis. *Plant Cell* **27**, 649–662, doi:10.1105/tpc.114.132274 (2015).

REVIEWERS' COMMENTS

Reviewer #1 (Remarks to the Author):

In this second revision of their manuscript, the authors have answered to all of my previous concerns. I thank them for taking my comments into consideration.

Reviewer #3 (Remarks to the Author):

The authors have addressed my remaining concerns and I do support publication of the manuscript in its current form.

REVIEWERS' COMMENTS

Reviewer #1 (Remarks to the Author):

In this second revision of their manuscript, the authors have answered to all of my previous concerns. I thank them for taking my comments into consideration.

Re: Thanks. We appreciated your excellent suggestions for making this paper and would like to give our formally acknowledgement for your contributions.

Reviewer #3 (Remarks to the Author):

The authors have addressed my remaining concerns and I do support publication of the manuscript in its current form.

Re: Thanks. We appreciated your excellent suggestions for making this paper and would like to give our formally acknowledgement for your contributions.